# EatA mediated degradation of intestinal mucus is species-specific and driven by MUC2 structural features

Sergio Trillo-Muyo [1,5], Brendan Dolan [1,5], Frida Svensson[1], Tim J. Vickers[2], Liisa Arike [1], Maria-Jose García-Bonete [1], Jenny K. Gustafsson [3], Mathias I. Nielsen [4], Hans H. Wandall [4], James M. Fleckenstein [2], Gunnar C. Hansson [1] & Sjoerd van der Post [1] ✉

Enterotoxigenic *Escherichia coli* (ETEC) infections are a leading cause of diarrheal illness, responsible for an estimated 100,000 deaths annually. ETEC pathogenesis is driven by various virulence factors, including toxins, adhesins, and noncanonical factors such as the protease EatA. The first line of host defense against intestinal pathogenic bacterial infections is the protective intestinal mucus layer. Here, we demonstrate the mechanism by which EatA degrades the core mucus component MUC2, thereby facilitating access to the epithelial cell surface and promoting infection. We identify the specific cleavage site region localized at the C-terminal of MUC2. EatA's protease activity depends on the interaction between two distinct, uniquely spaced domains in human MUC2, which defines species specificity. We confirm this using a novel transgenic mouse model exclusively expressing human MUC2, which allows us to study the role of the mucus layer in the infection by human intestinal pathogens. These findings highlight how ETEC is adapted to specifically degrade the mucus layer of its human host.

Enterotoxigenic *Escherichia coli* (ETEC) infections are one of the major causes of diarrheal illness in low- and middle-income countries[1]. While most of the population in endemic regions encounters ETEC, its effects are particularly severe for vulnerable groups. In particular, children under the age of five are especially vulnerable to acute symptomatic infections[2]. ETEC infections in young children are also significantly associated with long-term sequelae, including malnutrition and stunted growth[3]. ETEC infections are transmitted through contaminated food and water, and subsequently colonize the ileum by adhering to glycolipids and glycoproteins on the epithelial surface, enabling the efficient delivery of heat-labile and/or heat-stable enterotoxins[4,5]. Activation of cyclic nucleotide pathways by these enterotoxins leads to modulation of cellular ion channels and disruption of water and electrolyte balances, resulting in diarrhea[6]. Beyond the adhesins and toxins, additional colonization factors contribute to ETEC virulence[7]. These include secreted proteases belonging to the serine protease autotransporters of Enterobacteriaceae (SPATE) family[8]. Importantly, SPATE expression by diarrheagenic *E. coli* pathovars, including enteroaggregative *E. coli* (EAEC) and ETEC, has been associated with virulence in young children[9–11].

The main physical barrier in the intestine preventing direct interaction between pathogenic microorganisms and the host epithelium is the mucus layer[12]. As proximity to the intestinal epithelium is important for adhesion and subsequent toxin delivery, bacteria equipped with a strategy to overcome the mucus layer may exhibit enhanced virulence. We have previously determined that ETEC

[1]Department of Medical Biochemistry and Cell Biology, University of Gothenburg, Gothenburg, Sweden. [2]Department of Medicine, Division of Infectious Diseases, Washington University School of Medicine, Saint Louis, MO, USA. [3]Department of Physiology, University of Gothenburg, Gothenburg, Sweden. [4]Copenhagen Center for Glycocalyx Research, ICMM, University of Copenhagen, Copenhagen, Denmark. [5]These authors contributed equally: Sergio Trillo-Muyo, Brendan Dolan. ✉e-mail: Sjoerd.van.der.Post@medkem.gu.se

secretes EatA, a member of the SPATE family that can degrade the MUC2 mucin, the main structural protein component of the intestinal mucus layer[13,14]. The MUC2 mucin is characterized by its central mucin domains, which are composed of multiple repeated sequences composed almost exclusively of proline, serine and threonine (PTS-tandem repeats) that serve as attachment sites for *O*-glycans essential for providing mucus its gel-like properties[15]. The central mucin domains are flanked by two terminal protein assemblies that are highly stabilized via disulfide bridges and form large oligomeric net-like structures through additional intermolecular disulfide bonds. These distinct features render the protein highly resistant to degradation by members of the commensal microbiota, and only pathogens, including bacteria and parasites, have been demonstrated to produce proteases that act on MUC2 and the oligomeric mucus gel[16–18]. This highlights the necessity of intestinal pathogens to adapt strategies to degrade the mucus to facilitate access to the underlying tissue to increase virulence.

In the present study, we provide a detailed mechanistic insight into how the ETEC protease EatA degrades the intestinal mucus layer. Identifying the proteolytic cleavage site and demonstrating the importance of the MUC2 protein structure for substrate recognition, a process dependent on the interaction between multiple domains. Using structural modeling and a novel transgenic human MUC2 expressing mouse model, we demonstrate how variations in the protein structure drive species specificity. Altogether, these data provide evidence that pathogenic *E. coli* have adapted to degrade the mucus specific to their host.

## Results

### The ETEC serine protease EatA acts on the C-terminal protein region of MUC2

Previous studies have demonstrated that EatA is proteolytically active on the MUC2 mucin[13]. To identify the region of MUC2 susceptible to cleavage we used recombinant proteins spanning different regions of the protein (Fig. 1A). When N- and C-terminal regions of MUC2 (hereafter referred to as MUC2N and MUC2C) were treated with EatA in an oxidizing environment and analyzed by SDS–PAGE under reducing conditions, activity was observed only on the C-terminal region (MUC2C). EatA reduced the apparent molecular weight from >250 to ~170 kDa, consistent with the generation of a large proteolytic fragment (Fig. 1B). The terminal regions of the MUC2 protein facilitate dimerization through intermolecular disulfide bonds. These dimers then assemble into large oligomeric structures, which are further stabilized internally by additional disulfide bonds[19–21]. To determine if EatA can dissociate dimerized MUC2N or MUC2C, the protease assay was performed under oxidizing conditions and analyzed by SDS–PAGE under non-reducing conditions. The results confirmed the resistance of the N-terminal, while in contrast, the C-terminal multimeric structure was degraded (Fig. 1C). To narrow down the region susceptible to proteolytic cleavage, an additional C-terminal protein (MUC2C-NC) was tested spanning the region between the von Willebrand CN domain (VWCN) and the cystine-knot (CK) at the C-terminus (Fig. 1A)[20]. Analysis by protein gel electrophoresis of the truncated C-terminus of MUC2 upon EatA treatment resulted in an unaltered band migration (Fig. 1D). This result indicates that the EatA cleavage site is before the VWCN domain. In support of this finding, an additional protein, MUC2C-N, was used. This protein was truncated at the C-terminus, containing only the N-terminus of the MUC2 C-terminal protein region of MUC2 (PTS3, VWCN, D4, and VWC1). Gel analysis revealed that EatA treatment resulted in a reduced band size, emphasizing that the approximate region of the cleavage site is in the PTS3-VWCN region (Fig. 1E). To confirm that the observed activity on MUC2C-N was specifically due to EatA and not a contaminating protease, a non-catalytic mutant of EatA was generated by substituting histidine 134 with an arginine. Additionally, the serine protease nature of EatA was validated

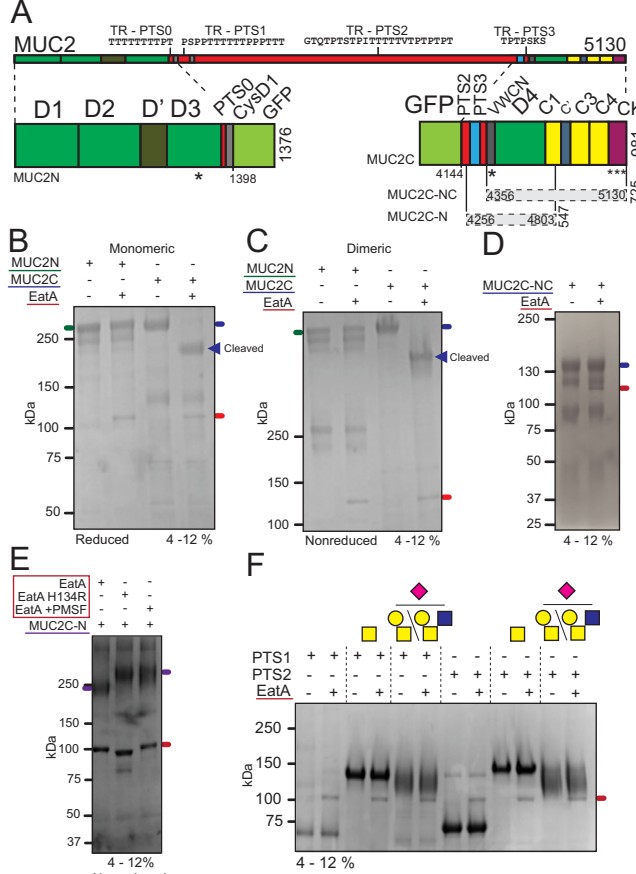

**Fig. 1 | The C-terminal protein region of MUC2 is susceptible to degradation by the protease EatA. A** Schematic representation of the domain organization of the MUC2 mucin and the recombinant fusion proteins used in this study. The depicted domains are von Willebrand assembly (D), Von Willebrand C-type (C), tandem repeat mucin domain (TR-PTS), cystine-knot (CK), CysD1 domain (CysD) and green fluorescent protein (GFP). The location of the intermolecular disulfide bonds responsible for dimerization are indicated with an asterisk (*). **B, C** SDS–PAGE analysis under non-reducing and reducing conditions of the recombinant MUC2N and MUC2C after EatA treatment visualized by Coomassie. **D** SDS–PAGE analysis and Coomassie staining of the truncated C-terminal protein MUC2C-NC treated with EatA. **E** MUC2C-N treated with active, non-catalytic mutant or inhibited EatA analyzed under non-reducing conditions by SDS–PAGE and visualized by Coomassie. **F** SDS–PAGE of the MUC2 PTS1 and PTS2 variants modified with single GalNAc residues or sialylated Core-1 or 2 oligosaccharides after treatment with the EatA protease. Representative qualitative gel electrophoresis images from two independent experiments are shown for (**B–F**) (*n* = 2). Colored lines underneath the protein names color-match with the lines on the gel images indicating the position of the specific protein.

by treatment with PMSF, a serine protease inhibitor. Neither the mutant EatA or PMSF-treated EatA were able to degrade MUC2C-N (Fig. 1E)[13]. The analyses were performed under non-reducing conditions, as the EatA and MUC2C-N protein are similar in size under reducing conditions.

The MUC2N and MUC2C-NC/MUC2C-N proteins lack the central mucin domain of the MUC2 mucin composed of multiple variable number tandem repeats (VNTR). These repeats are predominantly composed of proline, threonine, and serine (PTS) residues and are the main site of *O*-glycosylation. Protease activity was validated on two proteins composed of varying number of the PTS1 (nine repeats) or PTS2 (seven repeats) domains modified with either only GalNAc residues or extended to Core-1 or -2 structures with additional terminal sialylation (Fig. 1F)[22]. EatA treatment of the two PTS domains did not

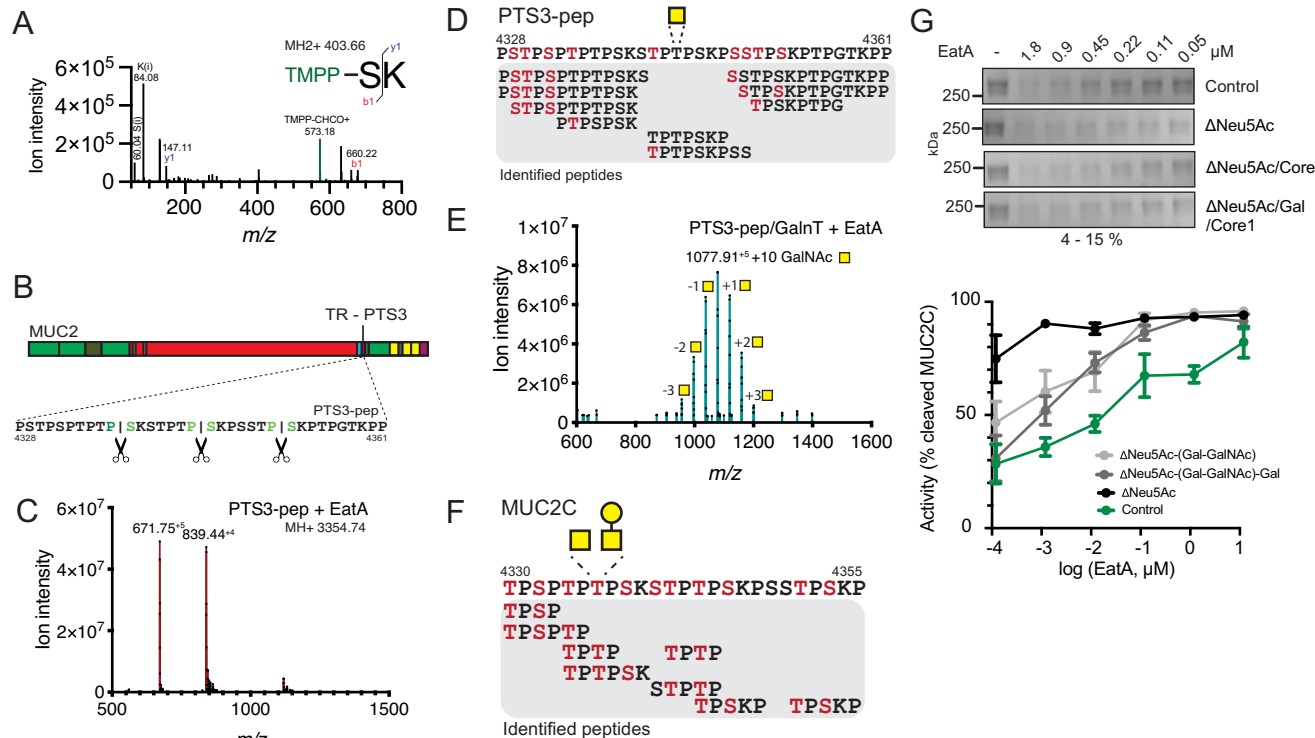

**Fig. 2 | Identification of the EatA cleavage site in the MUC2 C-terminal.**
**A** Fragmentation spectra of the neo N-terminal after EatA cleavage followed by trypsin digestion identified the amino acid sequence SK. The reporter ion at 573.18 $m/z$ indicates TMPP labeling, and the b-1 fragment the N-terminal serine, the peptide composition is further supported by the immonium (i) ions. **B** Overview of the EatA cleavage site in MUC2 and its reoccurrence in the PTS3 repeat. **C** Mass spectrum of the recombinant PTS3 spanning peptide after treatment with EatA. The MH+4 and MH+5 peaks represent the mass of the full-length 34-mer peptide at a mass of 3353.73 Da. **D** Glycopeptides identified by mass spectrometry of the recombinant GalNAc-tranferase modified PTS3-pep containing peptide, serines and threonines highlighted in red indicate residues that were found modified with GalNAc.

**E** Mass spectrum of the GalNAc-modified recombinant TR-PTS3 peptide after treatment with EatA. The number of GalNAc residues introduced was between 7 and 13 with a majority of ten modified sites indicated by the predominant peak at 1077.91 MH+5. **F** Glycopeptides identified by mass spectrometry spanning the TR-PTS3 region in recombinant MUC2C, serines and threonines highlighted in red indicate residues that were found modified with GalNAc or GalNAc-Gal. **G** Kinetics of MUC2C digestion by EatA after sequential deglycosylation with sialidases, galactosidase and endo-$N$-acetylgalactosaminidase as compared to control, determined by SDS–PAGE electrophoresis and represented as percentage cleaved mean ± SEM ($n = 3$).

result in degradation (Fig. 1F). These results indicate that EatA has proteolytic activity specifically targeted towards the MUC2 C-terminal in the region between the PTS2 and VWCN domain. Cleavage in this region results in the degradation of the complex oligomeric mucin structures.

## The EatA cleavage site is in the C-terminal PTS3 domain of MUC2

Proteases of the SPATE family have a broad substrate specificity, predominantly towards $O$-glycosylated secreted and cell surface proteins[23]. One of the first established substrates of SPATEs was IgA1, which can be cleaved in the hinge region between the Fab and Fc domains. The protease responsible for cleaving IgA1 was shown to be the immunoglobulin A protease (IgAP), which is secreted by various pathogens including *Haemophilus influenza*. When testing if IgA1 was an EatA substrate, we observed that EatA is not proteolytically active on IgA1 as demonstrated by gel electrophoresis (Supplementary Fig. S1A). The IgA1 hinge region is predominantly composed of serine and threonine residues and modified with three to six Core-1 $O$-glycans[24]. This region resembles the PTS repeats in MUC2, and a sequence similarity search against the human proteome with the IgA1 hinge region identified MUC2 as the protein with the highest local alignment, with 73% sequence similarity to the tandem repeat PTS3 region in the C-terminal region of MUC2 (amino acid 4329 to 4348) (Supplementary Fig. S1B). The high sequence similarity at the IgA1 cleavage site, coupled with the lack of EatA activity on IgA1, suggests that other regions of the molecule are important for substrate recognition.

To determine the cleavage site in the MUC2 C-terminus, we used mass spectrometry to detect the neo-N-terminus generated by EatA digestion of MUC2C. The method is based on α-amine specific labeling with the TMPP reagent, which retains the positive charge essential for peptide sequencing[18,25]. The EatA-treated MUC2C protein was labeled with TMPP prior to analysis by protein gel electrophoresis and staining with Coomassie. The 170 kDa MUC2C band was excised, and the protein was digested using trypsin. Resulting peptides were analyzed by mass spectrometry, and the TMMP-labeled dipeptide SK was identified as the neo-N-terminal formed after EatA cleavage (Fig. 2A). The SK motif appears three times within the PTS3 region, indicating that EatA acts on this region of MUC2. However, due to the repetitive nature of the sequence, the exact cleavage site could not be determined. (Fig. 2B). To refine mapping of the cleavage site, a recombinant non-$O$-glycosylated 34-mer peptide covering the cleavage site was synthesized corresponding to amino acid residues 4328 to 4362 (denoted as PTS3-pep). Incubation of the PTS3 peptide with EatA, and subsequent mass spectrometry analysis yielded only peaks representing the multiply charged intact peptide and no cleaved product (Fig. 2C). Together, these results localize the EatA activity to the PTS3 region but indicate that cleavage depends on determinants absent in the recombinant PTS3 peptide.

The PTS3 region of MUC2 is likely glycosylated; thus, the EatA activity may be dependent on the presence of $O$-glycans. To address this, we modified the PTS3 peptide using a combination of recombinant GalNAc-tranferases, introducing between 7 and 13 GalNAcs to the

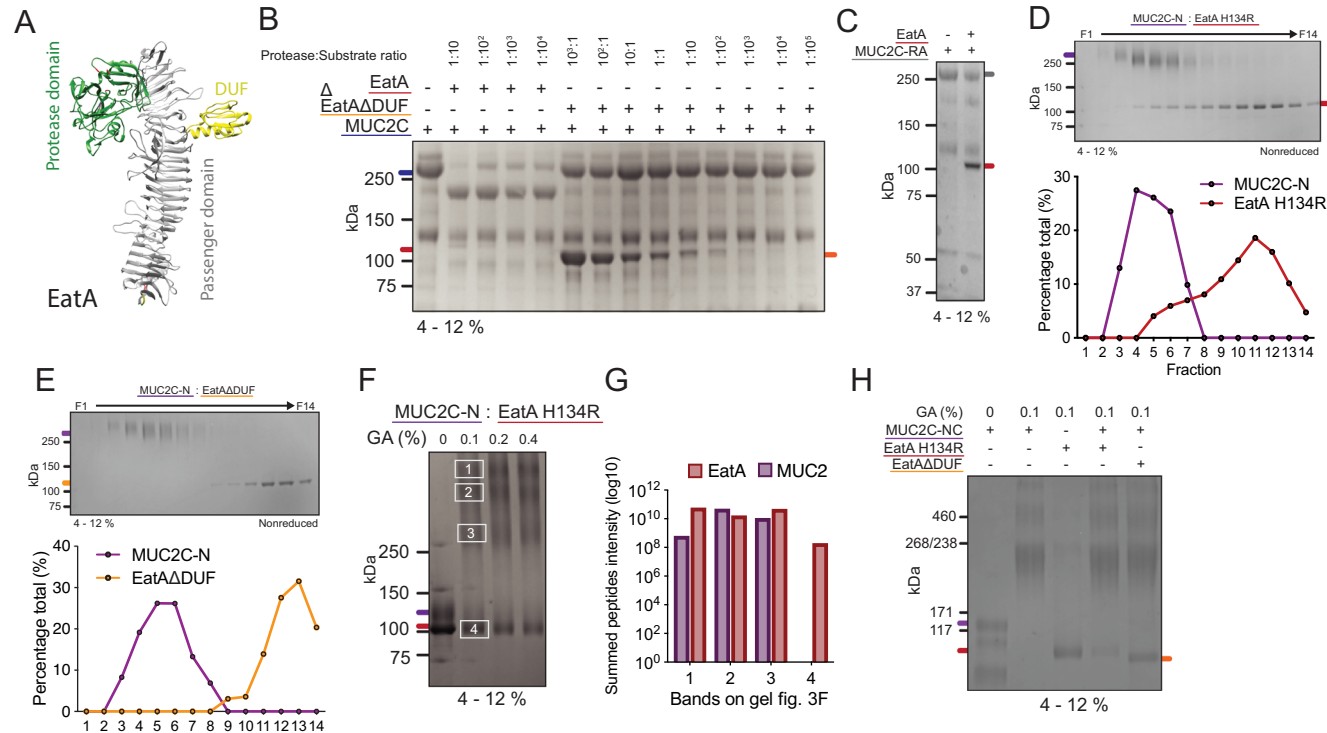

**Fig. 3 | MUC2 structural features are critical for recognition by EatA. A** Structure of the secreted passenger domain of EatA as predicted using AlphaFold2 (Uni-ProtID: Q84GK0). The serine protease domain in green, the domain with unknown function (DUF) in yellow and the passenger region in gray. **B** Protein gel electrophoresis of MUC2C treated at varying protease to substrate ratios of EatA or EatAΔDUF lacking a domain extending from the helical stalk region. **C** SDS−PAGE analysis of the EatA protease activity on unfolded, reduced, and alkylated MUC2C-N (MUC2C-RA). **D**, **E** Quantification of the proportion of MUC2C-N and EatA H134R or EatAΔDUF in each fraction (F1–14) after 1 h preincubation and subsequent size-exclusion chromatography. Distribution was determined based on band intensity after protein gel electrophoresis. **F** Protein crosslinking of MUC2C-N and EatA H134R with increasing concentrations of glutaraldehyde (GA). **G** Mass spectrometry analysis of the protein components of the bands after glutaraldehyde crosslinking. Proteins were digested using trypsin, and the data were represented as the summed peptide intensities for each protein in the sample (*n* = 1). **H** Protein crosslinking of MUC2C-NC lacking the cleavage site region and EatA H134R or EatAΔDUF using glutaraldehyde (GA). Representative qualitative gel electrophoresis images from two independent experiments are shown for (**B**, **C**, **H**) (*n* = 2). Colored lines underneath the protein names color-match with the lines on the gel image representing the position of the specific protein.

34-mer peptide (Fig. 2E). Glycosylated residues were localized by mass spectrometry after digesting the modified peptide with the *O*-glyco-protease OgpA (Fig. 2D and Supplementary Data 1)[26]. Treatment of the GalNAc-transferase modified peptide with EatA for up to 24 h did not result in any cleavage product (Fig. 2E). Thus, EatA activity on the PTS3 region alone is not dependent on *O*-glycosylation.

We next evaluated the role of *O*-glycosylation for EatA activity on the MUC2C protein rather than the PTS3 peptide. To assess whether the *O*-glycosylation of the recombinantly GalNAc-transferase modified PTS3 peptide reflects the native cellular state, we characterized the modified sites and *O*-glycan composition on MUC2C. Glycopeptides were generated by desialylation and digestion with OgpA and analyzed by mass spectrometry. The results showed that MUC2C was *O*-glyco-sylated at 10 serines and threonines by either GalNAc or GalNAc-Gal, confirming a similar site occupancy as observed for the glycosylated PTS3 peptide (Fig. 2F and Supplementary Data 2). As expected for proteins produced by CHO-K1 cells, only Core-1 glycans were observed on MUC2C, and site occupancy varied, reflecting the CHO-K1 cells' limited capacity to generate extended *O*-glycan structures[27]. Stepwise *O*-glycan removal from MUC2C using sialidases, galactosidase and endo-*N*-acetylgalactosaminidase did not prevent cleavage by EatA (Fig. 2G). Both partial and complete removal of the glycan residues enhanced the protease kinetics, specifically following desialylation, which is in line with sialic acid's role in protecting glycoproteins from proteases, as demonstrated for other SPATEs[28,29]. These results confirm that EatA cleaves in a region of MUC2 that is highly *O*-glycosylated,

but is not depending on *O*-glycans for activity or substrate recognition. Nevertheless, cleavage of MUC2C requires the intact protein, as the PTS3 peptide alone was not cleaved.

## EatA activity is dependent on the structure of MUC2

The inability of EatA to degrade the recombinant peptide spanning the identified cleavage site indicates that the protease depends on the secondary protein structure and folding of MUC2 for protease activity. In addition to the protease domain, EatA contains additional smaller domains extending from the long beta-helix region of the passenger domain, which have been demonstrated by us and others to be essential for substrate recognition[13,30]. The frequency and size of these domains varies between SPATE classes, indicating a specific role[31]. The largest domain with unknown function (DUF) lies within the stalk region of EatA and is a feature common to other members of the SPATE family (Fig. 3A). To determine the function of the previously identified DUF domain for protease activity on MUC2, an EatA mutant lacking the DUF domain was tested[13]. MUC2C was incubated with decreasing concentrations of EatA or EatAΔDUF and analyzed by gel electrophoresis (Fig. 3B). EatA was able to degrade MUC2C even at a 1:10,000 protease to substrate ratio, while EatAΔDUF was unable to degrade MUC2C, not even when using a large excess of the protease. This highlights that the DUF domain is essential for the recognition of MUC2.

To determine how important the structure of MUC2 is for sub-strate recognition, we unfolded the MUC2 by reducing all disulfide

bonds, followed by alkylation of the free sulfhydryl groups to prevent refolding. EatA was unable to degrade MUC2C after disrupting the disulfide bonds indicating that the protease is dependent on secondary protein structure in addition to the presence of a specific cleavage site (Fig. 3C). The interaction between the protein and the protease was demonstrated under native conditions by size-exclusion chromatography after incubation of MUC2C-N and the catalytically impaired H134R EatA mutant for 1 h prior to separation. The partial overlap between the elution of MUC2C-N and EatA indicates that the two proteins exist as a complex (Fig. 3D). In contrast, when the DUF domain was removed, the interaction was not observed (Fig. 3E). The elution profiles of the individually analyzed proteins show that the observed overlap between H134R EatA and MUC2C-N arises from their interaction. (Supplementary Fig. 2A, B). The size-exclusion chromatography results indicate an interaction between protease and substrate. To further demonstrate the direct interaction between MUC2 and EatA, the protein complex was chemically crosslinked using glutaraldehyde and analyzed by gel electrophoresis. The crosslinking formed protein complexes far exceeding the size of the MUC2C-N dimer observed around 250 kDa, indicating oligomerization between the two proteins (Fig. 3F). Mass spectrometry analysis of the protein composition of the four main bands observed on the gel confirmed the presence of both MUC2 and EatA in the three high molecular weight bands (Fig. 3G and Supplementary Data 3). To determine the interaction between EatA and MUC2 C-terminal beyond the PTS3 domain, we crosslinked the mutant protease and the MUC2C-NC protein composed only of the C-terminal beyond the cleavage site (Fig. 1A). We observed that the combination still resulted in a reduction of the EatA band around 100 kDa and an increase in higher molecular weight bands. While no crosslinking was observed in the protease lacking the DUF domain (Fig. 3H). This suggests that, in addition to the PTS3, the DUF domain is required for protease interactions between the MUC2 C-terminal region after the cleavage site, which includes the von Willebrand factor CN, D4, C1, and C3 domains and cystine-knot. These results demonstrate that coordinated interactions between distinct regions of EatA and MUC2 are essential for protease activity, involving the DUF domain on the stalk region of EatA and MUC2 domains C-terminal to the cleavage site.

## EatA selectively acts on the human intestinal mucus layer

The MUC2 mucin is the major structural component of the mucus layer covering the epithelial surface in the small and large intestine. By creating large oligomeric mesh-like structures, it limits the penetration of the commensal microbiota residing in the intestinal lumen[32]. The pathogenesis of ETEC depends on adherence to intestinal epithelial cells, followed by the secretion of enterotoxins, a process that is normally limited by the protective mucus layer. To determine if EatA can degrade the mucus layer to facilitate access in vivo, we applied our ex vivo mucus measurement setup[33]. Colonic tissue was mounted horizontally in a perfusion chamber kept at 37 °C. The epithelium was visualized using a DNA stain, and fluorescent beads in the size range of bacteria were applied apically to visualize the top of the mucus layer. The protease was added to the apical tissue surface, and the thickness of the mucus layer was monitored over time by confocal microscopy. Ex vivo mucus measurements of colonic tissue from wild-type C57BL/6N mice treated with EatA for up to 20 minutes had no effect on the thickness or penetrability of the mucus layer as indicated by the constant distance between the cell surface and the beads on top of the mucus (Fig. 4A, B and Supplementary Videos 1, 2). These results indicate that the mucus in mice is not degraded by EatA. Since mice are not susceptible to ETEC infection, we hypothesized that this is potentially due to an inability of ETEC to degrade the mouse Muc2 mucin. To address the role of species specificity for EatA substrate recognition, we generated a transgenic mouse model in which we introduced human MUC2. Human MUC2-expressing mice were crossbred with

Muc2 knockout animals to generate $MUC2^+/Muc2^{-/-}$ mice, which have a humanized mucus layer. Treatment of mucus in the $MUC2^+/Muc2^{-/-}$ mice with EatA resulted in a rapid degradation of the mucus layer, as shown by the decreased distance between the beads and the epithelial surface (Fig. 4C and Supplementary Fig. 3A). Beads were observed to settle in clusters and float away in the absence of the mucus layer (Supplementary Video 3). Mucus analysis without EatA addition demonstrates that the mucus of $MUC2^+/Muc2^{-/-}$ mice maintained its barrier function over time (Fig. 4D, Supplementary Fig. 3B, and Supplementary Video 4). Repeated ex vivo analysis corroborated that only the $MUC2^+/Muc2^{-/-}$ mucus is susceptible to degradation by EatA (Fig. 4E). To confirm that the transgenic mice generate a functional mucus layer that is maintained over time, we used an additional mucus measurement technique in which the thickness is measured using a penetrating needle while the mucus surface is visualized by beads[34]. Both initial mucus thickness and growth over time were similar in C57BL/6N and $MUC2^+/Muc2^{-/-}$ mice; however, only mucus produced by $MUC2^+/Muc2^{-/-}$ mice was degraded by EatA (Fig. 4F).

To further validate the EatA observations in the humanized mucus mouse model, we repeated the analysis using human colonic biopsies. Biopsies were obtained from control patients without a history of intestinal disease and analyzed in the same way as the mice. EatA treatment of the mucus resulted in degradation of the layer, while the mucus layer remained intact when treated with PMSF inhibited EatA (Fig. 4G–I and Supplementary Videos 5, 6). These results demonstrate that EatA selectively degrades the human MUC2 mucin, but not murine Muc2.

## MUC2/Muc2 PTS3 sequence variation dictates species-selective mucus degradation

The cleavage site region comparison between human MUC2 and mouse Muc2 shows a clear variation in both length and amino acid composition (Fig. 5A). The identified cleavage site in MUC2 is present twice in Muc2 with the same residues in the P1-P1' position (P/S) but without a lysine in the P2' position. The lack of a positively charged residue in this position potentially impairs substrate recognition. We assessed EatA activity on the mouse C-terminal of Muc2 using a recombinant protein spanning the same region of the protein as the human MUC2C-N protein (Fig. 1A). Western blot analysis targeting the N-terminal FLAG-tag demonstrates that only human MUC2 is degraded while mouse Muc2 remains intact (Fig. 5B). To determine if the mouse Muc2C-N binds to EatA we crosslinked the two proteins with increasing concentrations of glutaraldehyde. This combination resulted in a decreased EatA band intensity, indicating complex formation, which was dependent on the presence of the DUF domain (Fig. 5C). In addition to humans, pigs, cattle, sheep, and dogs are susceptible to ETEC infection, especially during the neonatal period[35]. By contrast, mice and rats are not. When comparing the amino acid sequences of the PTS3 domain in the affected species to that of rodents, the major variation was observed in the length of the region (Fig. 5D). While the terminal regions are generally conserved, an -18 amino acid insertion is present in both mouse and rat Muc2. These additional residues will affect the spatial arrangement of the cleavage site region and potential accessibility for the protease. These results indicate that EatA activity is tailored to degrade the mucus of its specific host species. Providing an insight into one of the determinants of why human colonizing ETEC strains are ineffective in mice.

## Model of the EatA binding to MUC2

To gain insight into the molecular mechanism of EatA-MUC2 recognition and cleavage, we performed docking simulations using a model based on the MUC2 C-terminal homodimer (vWCN−D4−VWC1) cryo-EM structure (PDB: 7QCN) and an AlphaFold2 model of EatA (UniProt:Q84GK0)[20]. The AlphaFold2 EatA model showed very high confidence except for the DUF domain, the region suggested to drive

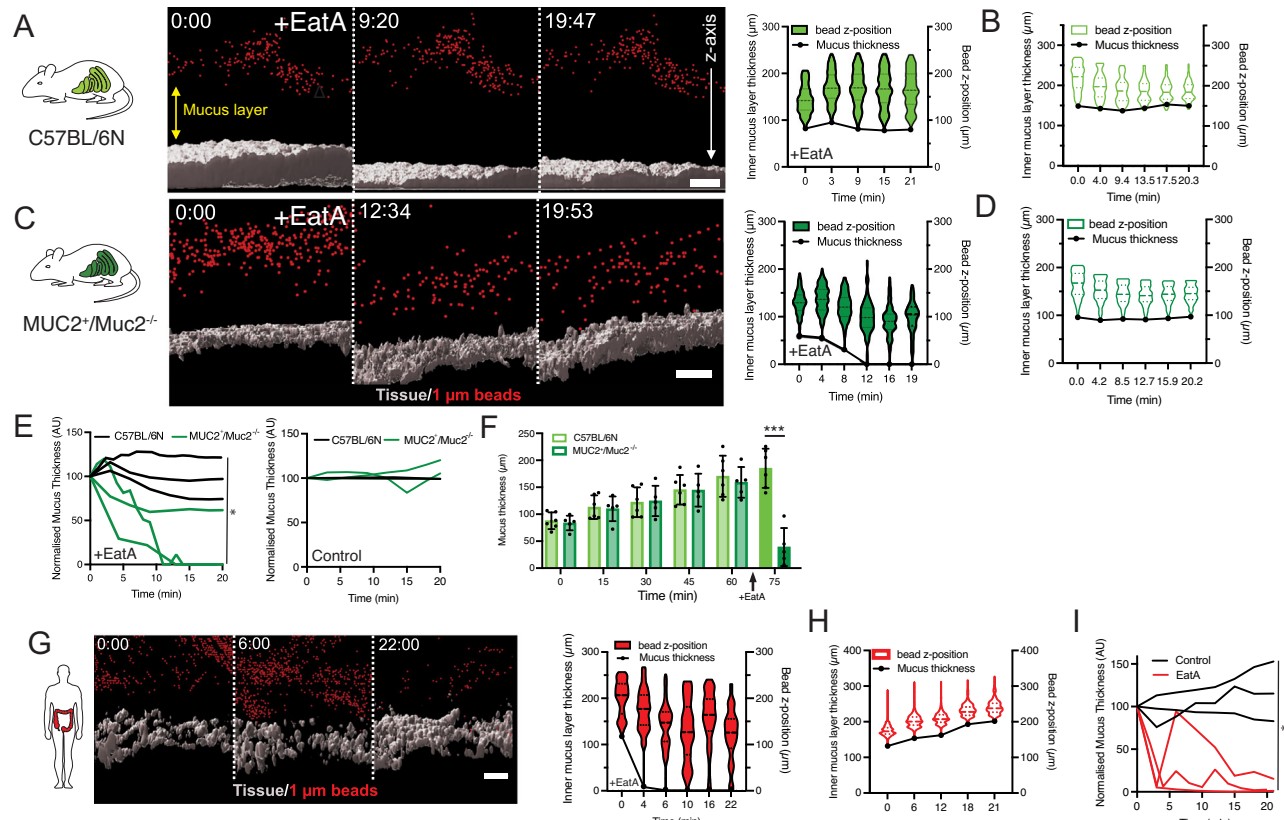

**Fig. 4 | EatA specifically degrades the human colonic mucus layer.**
**A** Representative z-stack projection of ex vivo mucus penetrability analysis in the colon of wild-type C57BL/6N mice upon EatA treatment over time, showing the bead position (red) in relation to the tissue (gray). The violin plot shows bead frequency distribution in relation to tissue surface over time. Dashed black lines indicate the median, and solid lines the center quartiles. **B** Control non-protease-treated wild-type C57BL/6N mucus penetrability analysis over time.
**C** Representative z-stack projections over time of ex vivo mucus penetrability analysis in the colon of MUC2+/Muc2−/− mice upon EatA treatment. Violin plots show mucus penetrability in response to EatA treatment at each timepoint **D** Control non-protease treated MUC2+/Muc2−/− mucus shows normal penetrability over time.
**E** Combined data of replicated analysis of the penetrability assay on wildtype and

MUC2+/Muc2−/− upon EatA treatment (n = 3). **F** Comparison of colonic mucus layer thickness and mucus growth rate over time in C57BL/6 N (n = 6) and MUC2+/Muc2−/− (n = 5) mice, and the effect of EatA, error bars represent standard deviation. Data were presented as the mean with standard deviation. **G** Analysis of the mucus penetrability in human colonic biopsies and the response upon EatA treatment. **H** and in response to PMSF inhibited EatA. **I** Combined data of replicated analysis of the normalized penetrability assay on patient biopsies, mucus treated with EatA or the inactivated protease (n = 3). The silhouettes used in figure (**A**, **C**) were obtained from https://www.phylopic.org/ and are used under CC0 1.0 Universal public domain license. Scale bars are 50 μm. *p < 0.05, **p < 0.01 and ***p < 0.001 as determined by two-sided Student t-test at final timepoint (**E**, **I**), and by two-way ANOVA between mouse models at each timepoint for (**F**).

the interaction between the two proteins. The DUF prediction presented low confidence areas in the regions connecting to the passenger domain. The predicted alignment error (PAE) between the DUF domain and the rest of the molecule was high, indicating low confidence in the relative position of the domain, typical for flexible regions. Consequently, the docking simulations were conducted using the isolated DUF domain first, then extended with the remaining part of EatA to generate a model of the full complex with the MUC2C dimer. Models with major steric impediments were removed, whereas minor clashes were tolerated and subsequently relaxed by (5 ns) molecular dynamics simulation. The best-scoring complexes showed virtually the same interaction mechanism, with the DUF domain inserted into the cavity formed by the two MUC2 VWCN-D4 chains of the homodimer (Fig. 6A, B). This region of MUC2 is not protected by *N*- or *O*-glycans that could affect the dimer formation and present a hydrophobic region between the VWCN(B) and the VWD(B) domains, where the hydrophobic residues from the tip of the DUF domain are predicted to interact. Notably, the N-terminus of the VWCN from chain A of the MUC2 dimer points towards the catalytic domain at ~45 Å from the catalytic serine. Consistent with this arrangement, the EatA-MUC2 interface closely resembles the IgA1 interface described for IgA protease from *Haemophilus influenzae* (Fig. 6C)[36].

The MUC2 PTS3 was modeled by extending it from the VWCN domain until it reached the EatA active site based on the observation that PTS domains extend linearly due to the extensive *O*-glycosylation (Fig. 6A)[37]. According to the model, the most C-terminal of the three putative cleavage sites (Pro[4352]-Ser[4353]) is too close to the VWCN domain to be cleaved. The first cleavage site (Pro[4337]-Ser[4338]) cannot be excluded, but it is the second site (Pro[4344]-Ser[4345]) that matches the distance to the active site. Therefore, the Pro[4344] was manually placed in the S1 pocket and Ser[4345] in S1′ (Fig. 6D, 0 ns). *O*-glycans were added to all serines and threonines in PTS3, including Ser[4345] in P1 position, and the model was analyzed by molecular dynamics to evaluate the docking prediction, determine the stability of the complex, and to improve the model. Obvious interactions between glycans and the protease domain were not observed, indicating that glycans near the cleavage site are tolerated without compromising activity. The molecular dynamics simulation disclosed that the interaction between the catalytic domain and the PTS3 is mainly stabilized by Lys[4346] in P2′, forming a salt bridge with Asp[120] in S2′ (Fig. 6D, 50 ns). Homology modeling of the mouse PTS3 using the human structure as a template indicates that all lysines were positioned too far from the active site (Fig. 6D, 0 ns). Mouse Asp[3794] occupied a position equivalent to human K[4345] and was therefore modeled inside the S2′ pocket. However, the MD simulation

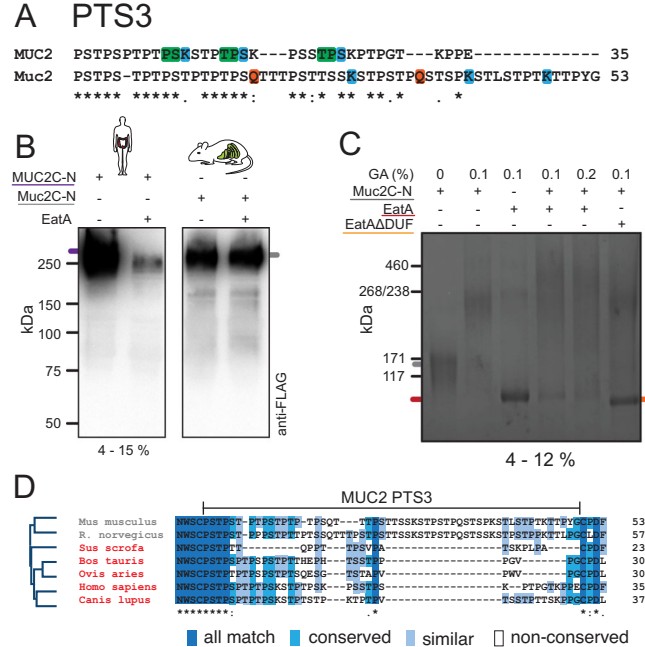

**Fig. 5 | Variation in the PTS3 sequence determines species specificity.**
**A** Alignment of the PTS3 region of mouse and human MUC2. The EatA P1-P1′ cleavage site position is indicated in green, and lysine and glutamine residues in blue and red. **B** Western blot analysis of the human MUC2C-N and mouse Muc2C-N protein constructs treated with EatA and probed for the presence of the N-terminal FLAG-tag (*n* = 2). **C** Protein crosslinking of Muc2C-N and EatA by using increasing concentrations of glutaraldehyde (GA) (*n* = 2). **D** Phylogenetic tree and multiple sequence alignment of the MUC2 PTS3 region displaying species unaffected (gray) and affected (red) by ETEC. The silhouettes used in figure (**B**) were obtained from https://www.phylopic.org/ and are used under CC0 1.0 Universal public domain license.

showed that after 10 ns, that Asp$^{3794}$ had moved out of the pocket and instead pointed toward the solvent (Fig. 6D, 10 ns).

The EatA-MUC2 model only suffered small rearrangements during the simulation; the interaction surfaces involving the DUF, passenger and protease domains were preserved (Supplementary Fig. 4A, B). The current model thus supports EatA recognition of a lysine in position P2′ and cleavage of the PTS3 not closer than ~15 amino acids N-terminally of the VWCN domain in an *O*-glycosylation-independent manner (Supplementary Video 7).

## Discussion

Enteric pathogens need to overcome the largely impenetrable mucus layer to engage with the intestinal epithelium[38]. Here, we demonstrate that ETEC can degrade the mucus layer utilizing the highly specialized SPATE protease EatA that directly acts on the main protein component of intestinal mucus, the MUC2 mucin. MUC2 forms oligomers that are highly resistant to proteolysis in the hostile environment of the intestinal lumen, leaving only a few regions susceptible to protease cleavage, which would result in disruption of the mucus gel[17,18]. The EatA cleavage site is localized in a mucin domain in the C-terminal part of the protein, closely resembling the central mucin domain, which has been considered to be proteolytically resistant due to its extensive *O*-glycosylation. Various *O*-glycoproteases acting on mucin domains in other proteins have been identified but none has been demonstrated to actively degrade MUC2, in contrast EatA is not dependent on glycosylation but tolerates glycans in proximity, as supported by molecular dynamics analysis[39–41]. The recombinant proteins used in this study were produced using CHO-K1 cells expressing only a subset of glycosyltransferases, limiting the *O*-glycosylation to sialylated Core-1

structures[18]. *O*-glycan structures identified on MUC2 in the human intestine are predominantly based on Core-3 and far more branched and extended[42]. We demonstrate that EatA is active on both recombinant and intestinal MUC2, highlighting that the core structure or the potential proximity of extended *O*-glycan structures does not affect EatA activity. Although *O*-glycans on mucin domains are dense and heterogeneous, limiting site-specific recognition, EatA effectively overcomes this barrier.

EatA is unable to act solely on a synthetic peptide covering the cleavage site in the PTS3 region alone, suggesting that additional binding sites are essential for stabilizing the substrate-enzyme complex. Previously, a putative binding domain (DUF) was identified to be essential for MUC2 recognition and degradation[13], which we here confirm binds with high specificity to a cleft C-terminally of the PTS3 in between the two VWD4 domains of the MUC2 homodimer. Importantly, proteolysis depends on dimerization of the MUC2 C-terminus, as the DUF domain engages the interface formed by both chains. In the absence of the DUF, no cleavage activity was observed, and binding to MUC2 was reduced, highlighting the importance of the DUF domain for initiating interaction. DUF domains are present in many members of the SPATEs family, and the domain is the likely key determinant of substrate specificity[43]. Earlier assumptions suggested species tropism of enteric pathogens to be dependent on the presence of the receptors for the enterotoxin and host-specific adhesins[35]. The lack of EatA activity on mouse mucus indicates that species variation in the MUC2 sequence also determines host selection. During infection, mucus is one of the barriers ETEC is required to overcome, and differences in mucins could therefore be considered a major contributor to host selection. Despite DUF-dependent binding to mouse Muc2, cleavage did not occur, supporting the concept that variation in the sequence of the PTS3 region determines specificity. Molecular dynamics analysis of the mouse PTS3 domain in the active site of EatA highlights the necessity for the lysine in the P1′ position for stabilizing the substrate. The absence of the lysine around the cleavage site region at the correct distance in mouse Muc2 highlights the limitations of conventional mice as models for pathogenic *E. coli* infections[44]. As ETEC lacks a suitable murine model, *Citrobacter rodentium* is commonly used as an alternative, expressing the Pic SPATE implicated in colonization[45]. *Citrobacter rodentium* infection of Muc2$^{-/-}$ mice is lethal, underscoring the crucial role of the mucus layer[46]. Pic is similar in domain organization to EatA, with a slight variation in the active site and spatial arrangement of the domains along the autotransporter domain to potentially facilitate selective Muc2 cleavage.

The high level of substrate specificity among SPATEs suggests that previous studies relying on crude mucus extractions from non-host species provide limited insights into the actual substrate, the mode of interaction, and its function in vivo. To overcome this limitation, we generated a transgenic model selectively expressing human MUC2, which have a normal functional mucus layer under baseline conditions. EatA treatment efficiently resolved the mucus from transgenic mice at a similar rate to that seen in human colonic biopsies, we choose to perform this analysis on intact colonic tissue for the purpose of visualization[34]. The colonic mucus layer shares protein composition and *O*-glycosylation with the small intestine, but is thicker, providing superior visualization of mucus degradation[47]. This novel mouse model has the potential to be used to systematically evaluate the contribution of microbial mucus-degrading proteases during infection of not only ETEC but also other human intestinal enteric pathogens. The necessity to develop a strategy to overcome the mucus layer suggests that many SPATEs are tailored to degrade MUC2, including SepA from *Shigella flexerni* and Pic from enteroaggregative *E. coli*[48,49]. This shared virulence axis among related *E. coli* pathotypes offers a potential avenue for the rational development of broadly protective vaccine strategies designed to block the initial binding to MUC2, thereby preventing mucus degradation and colonization[14]. Progress

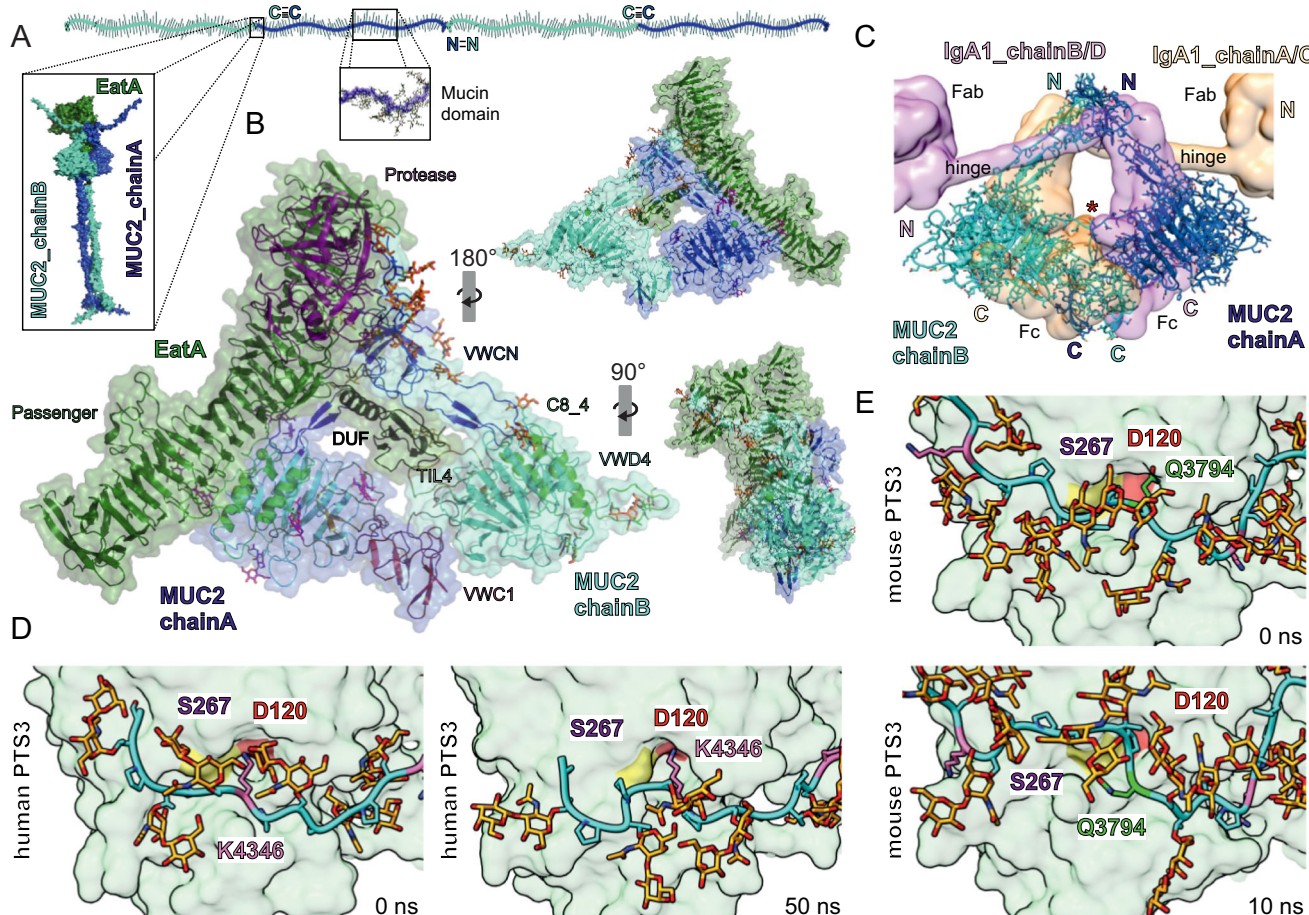

**Fig. 6 | EatA-MUC2 docking model.** EatA is shown in green, MUC2 chain A in blue and MUC2 chain B in cyan. **A** Schematic overview of an oligomeric MUC2 filament drawn to scale. The N- and C- termini where the disulfide-mediated oligomerization occurs are annotated. The predicted complex between the dimeric MUC2 C-terminal region and EatA is shown as an inset. **B** EatA-MUC2 docking model after 50 ns molecular dynamics (MD) simulation, surface (excluding glycans) and cartoon. Chains A and B of the MUC2 homodimer are labeled A and B, glycans on chain A are shown in magenta and those on chain B in orange. In the central figure cartoon representation, the different protein domains are shown in different colors: EatA Protease domain in magenta, Passenger domain in green and DUF in black; and MUC2 VWCN in blue, VWD4 in cyan, C8_4 in green, TIL4 in orange and VWC1 in red. The complex is turned clockwise by 90˚ on the y-axis in the bottom figure and 180˚ in the upper one. Both are reduced by 50% in size with respect to the central figure. **C** MUC2-IgA1 (PDB: 1IGA) superposition. IgA chain A and C surface is shown in light orange, and chain B and D in pink. MUC2 is represented by cartoons and sticks. IgA1 and MUC2 *N-* and *C-* termini are labeled, and IgA domains are specified. The IgA region predicted to interact with IgAP is marked with a red star. **D** EatA active site with human or **E** mouse MUC2 PTS3 detailed interaction before (0 ns) and after (50 ns human PTS3 and 10 ns for mouse PTS3) MD simulation. *O*-glycans are shown in orange, and the PTS3 main chain in cyan. Only the surface representation of EatA is shown. For human PTS3, the lysine in the P1' positions is labeled and marked in pink. In mouse PTS, all lysines (pink) and Q3794 (green) are marked. The S1 and S2' pockets are annotated, D120 in S2' is colored in red, and the catalytic serine (S267) in yellow.

toward such vaccines requires mapping the regions that mediate substrate recognition, so that inhibitory epitopes can be prioritized for immunogen design.

In conclusion, we present a mechanistic insight into the degradation of MUC2 by the ETEC protease EatA. We localized the cleavage site to a susceptible region of MUC2 that mediates mucus degradation and show that interspecies variation in this region underlies host selectivity. Protease activity in the unstructured region of MUC2 is dependent on the initial binding to a von Willebrand domain downstream of the cleavage site to stabilize and direct the flexible region of MUC2 to the catalytic site. The combined results underscore that mucus degradation requires highly specialized proteases, whose expression appears restricted to intestinal pathogens, highlighting a potential strategy to prevent ETEC infection.

## Methods

### Mouse and human subject details

All experimental procedures involving mice were performed in accordance with protocols approved by the laboratory animal ethics

committee guidelines at the University of Gothenburg (ethical permit: 2285/19). Mice were housed in a specific pathogen-free facility with access to standard chow and water *ad libitum* with a 12 h light/dark cycle, maintained at 22 °C ± 1 °C at 40−70% humidity. All mouse strains used were on a C57BL/6N background originating from Taconic. The Muc2 knockout strain has been previously described[50]. The MUC2 transgenic mice were generated via a pronuclear injection of a BAC clone (RP13-870H17) containing a segment of chromosome 11p15.5, including MUC2 and MUC6[51]. The *MUC6* gene was silenced by introducing a TetR cassette[52]. The mice expressing human MUC2 were crossbred with the Muc2 knockout to generate the MUC2⁺/Muc2⁻/⁻ strain. All animals used for experiments were in the age range between 8 and 12 weeks, and both males and females were included. Approval for the study of patient intestinal biopsies was granted by the human research ethical committee at the Sahlgrenska University Hospital, Gothenburg and written informed consent was obtained from all study subjects. Biopsies were obtained from the distal colon of patients undergoing colonoscopy with no history of inflammatory bowel disease. Only macroscopically normal tissue was included from patients

undergoing cancer screening due to occult blood in stool without signs of intestinal disease.

## Expression vectors, constructs, and recombinant proteins

The MUC2 N-and C-terminal recombinant proteins were expressed in CHO-K1 cells (CCL-61, ATCC) and purified from spent media by size-exclusion chromatography on a Superose 6 HR 10/30 column[53,54]. Expression and design of the truncated MUC2C protein MUC2C-CN is fully described in detail[20]. The pcDNA3.1(+) plasmid vector for the MUC2C-N construct was composed of an N-terminal Igk signal sequence followed by a HIS-tag, a FLAG-tag site and the MUC2 sequence between amino acid 4256 and 4803 (GenBank: AZL49145.1) with the cysteine residue at position 4786 mutated to an alanine. The mouse Muc2C-N vector was composed similar to the MUC2C-N including the Muc2 sequence between amino acid 3691 and 4248 (UniProt: Q80Z19 v2023_01) with the cysteine residue at position 4231 mutated to an alanine. CHO-S cells (R80007, Life Technologies) cultured in serum-free FreeStyle™ CHO growth medium (Gibco) were used for transient protein expression for 72 h. The spent media was concentrated, and buffer exchanged by tangential flow filtration and the protein was subsequently purified by immobilized metal affinity chromatography on a 1 mL HisTrap™ HP column (Cytiva) using an Ettan LC system (Amersham Bioscience). The eluted protein was concentrated and buffer exchanged to 20 mM HEPES, 150 mM NaCl, pH 7.4 using 50 kDa molecular weight cut-off filters (Vivaspin, Sartorius). The mucin tandem repeats VNTR-PTS1 (nine repeats, position in MUC2: 1406–1554) and VNTR-PTS2 (seven repeats, position in MUC2: 1916–2064) were expressed and purified from genetically-engineered HEK293 cells allowing the control of the addition of $O$-glycan modifications, limited to Tn (KO C1GALT1) or STn antigen (KO COSMC / KI ST6GALNAC1) as described in detail[22]. EatA and the non-catalytic H134R mutant were cloned into a pBAD vector, expressed in *E. coli* and purified from concentrated spent LB media by ion exchange chromatography followed by size-exclusion chromatography as described in detail[14]. Design and purification of the EatA protein in which the DUF domain spanning amino acid residue 514 to 616 was replaced by eight HIS residues are described in detail previously[13]. The 34-mer peptide spanning the VNTR-PTS3 between the amino acid 4328 to 4362 of MUC2 was synthesized (JPT Peptide technologies) and $N$-acetylgalactosamine modification were introduced using recombinant GalNAc-Ts transferases 2, 3, and 7[55].

## Protein gel electrophoresis and Western blot analysis

MUC2C was deglycosylated with β1-3 Galactosidase (P0726, New England Biolabs), α2-3,6,8 Neuraminidase (P0720, New England Biolabs) and $O$-glycosidase (P0733, New England Biolabs) either as a single reaction or sequentially. All reactions were set up according to the manufacturer's instructions without the addition of denaturing buffer and performed overnight at 37 °C. For the protease assays, recombinant MUC2 was combined with EatA and incubated at 37 °C for 1 h on a thermal shaker (300 rpm) if not otherwise stated. The protease activity was quenched by the addition of SDS–PAGE sample buffer (4% SDS, 125 mM Tris-HCl, pH 6.8, 30% (v/v) glycerol, 5% (v/v) bromophenol blue, supplemented with 200 mM DTT for reducing conditions) and heated for 5 min at 95 °C. Protein gel electrophoresis was performed on gradient NuPAGE Bis-Tris mini gels (Invitrogen) with a 4–12 or 4–15% polyacrylamide gradient and stained by Coomassie (Imperial, Thermo). The protein molecular weight marker used was either PageRuler pre-stained (26619, Thermo Fisher), Dual Color (1610374, Bio-Rad) or HiMark pre-stained (LC5699, Invitrogen). Western blot analysis was performed after transfer to PVDF by electroblotting for 10 min at 1.3 A/25 V (Trans-blot Turbo, Bio-Rad). Blots were blocked with 5% BSA in PBS-T for 1 h, probed with anti-FLAG antibody M2 (1:1000, F3165, Sigma, validated for western blot) for 2 h, and detected with anti-mouse IgG1-HRP (1:2000, 4030-05, Santa Cruz). Proteins on

blots were visualized by chemiluminescence after the addition of ECL ultra western HRP substrate (Millipore) and the band intensities were analyzed using Image Lab (v6.1, Bio-Rad).

## Protein interaction analysis

To determine complex formation, the non-catalytic EatA variant H134R or EatA lacking the DUF domain were incubated with MUC2C-N at an equal molar ratio for 1 h at 37 °C. The reaction mixture was applied to a Superdex 200 size-exclusion column (Cytiva) and eluted isocratically with 20 mM Tris, 150 mM NaCl, pH 7.4, monitoring UV at 215 and 280 nm using an Ettan LC system (Amersham Bioscience). Eluting fractions were analyzed by SDS–PAGE as described. For chemical protein crosslinking analysis, EatA H134R was incubated with MUC2C-N or Muc2C-N as above at a concentration of 0.5 μg/μL. Glutaraldehyde solution was added to a final concentration of 0.1, 0.2, or 0.4% and the reaction was terminated after 20 min by the addition of Tris buffer to a final concentration of 50 mM. Crosslinked protein samples were analyzed by SDS–PAGE and visualized by Coomassie (24615, Thermo).

## Sample preparation and mass spectrometry analysis

Trimethoxyphenyl phosphonium (TMPP) protein labeling was performed for the identification of the newly generated MUC2C-N-terminus after EatA cleavage. The protease reaction was quenched by the addition of 1 mM phenylmethylsulfonyl fluoride, the protein solution was titrated to pH >8 and <8.5 with $Na_2HPO_4$ and TMPP was added to a final concentration of 100 μM. The reaction was incubated at room temperature for 1 h and subsequently quenched by the addition of 0.1 M hydroxylamine and analyzed by SDS–PAGE. To determine the protein composition and identify TMPP or $O$-glycan modified sites, bands of interest were excised from the gel and destained, proteins were reduced, alkylated, and digested using trypsin (V5111, Promega) overnight at 37 °C, as described in detail[18]. For glycopeptide analysis, the protein was first desialylated with a combination of two broad-specific neuramidases (SialEXO, Genovis) and digested into peptides with a Core-1 specific glycoprotease (OpeRATOR, Genovis) in 25 mM ammonium bicarbonate, incubated overnight followed by trypsin for 2 h, all at 37 °C. The extracted peptides were dried under vacuum and dissolved in 0.2% trifluoroacetic acid and analyzed by mass spectrometry. Mass spectrometric analysis was performed by LC/MS/MS using an Easy-nLC 1200 (Thermo) coupled to a HF-X Orbitrap mass spectrometer (Thermo). Peptides were separated using in-house packed columns (150 × 0.075 mm) packed with Reprosil-Pur C18-AQ 3-μm particles (Dr. Maisch). Peptides were separated using a 5 to 35% gradient (A 0.1% formic acid, B 0.1% formic acid, 80% acetonitrile) over 30 minutes. Full mass spectra were acquired over a mass range of minimum 350 $m/z$ and maximum 1600 $m/z$, with a resolution of at least 60,000 at 200 $m/z$. The 12 most intense peaks with a charge state ≥2–5 were fragmented by higher-energy collision dissociation (HCD) with a normalized collision energy of 27%, and tandem MS was acquired at a resolution of 17,500 and subsequently excluded for selection for 20 s. Peaklists were generated from raw mass spectrometry data using MSConvert (v3.0) and searched using Mascot (v2.2.04, Matrix Science) for peptide identification and Byonic (v2.10.21, Protein Metrics) for glycopeptide identification. Mascot was used for standard protein identification with the following settings: fixed modification carbamidomethylation (C), variable modification oxidation (M), enzyme trypsin with a maximum number of two missed cleavages, a mass tolerance of 10 ppm for precursor mass and 20 ppm for fragment ions. For newly generated N-termini identification, the search settings were adapted to include TMPP (N-terminal, Y and K) and semi-trypsin for enzyme specificity. All peptide searches were performed against an in-house protein database (Mucin DB v2), including the recombinant proteins and EatA, only peptides at 1% FDR were considered. Byonic parameters were set to semi-specific for C-terminal cleavage trypsin (KR) and N-terminal OpeRATOR (ST) with a maximum number of missed

cleavages up to two, mass tolerance of 10 ppm for precursor mass and 10 ppm for fragment ions. The searches were performed against the mucinDB, and a list of 70 common human O-glycans was selected as the set of considered glycan modifications; only glycopeptides and peptides at 1% FDR were considered. The unmodified or GalNAc-modified 34-mer PTS3 peptide (JPT Peptide Technologies) was dissolved in PBS and further diluted with 0.2% trifluoracetic acid to 2 μM for analysis by LC-MS/MS using the same instrument settings as described. Protease cleavage was validated by combining 100 μM PTS3 with up to 1 μg EatA, incubated at 37 °C overnight and up to 24 h. The reaction was quenched by the addition of 0.2% trifluoracetic acid and analyzed by LC-MS/MS.

### Ex vivo analysis of EatA mucus degradation

Ex vivo mucus analysis[34,56] of EatA activity was performed as following after euthanasia, mouse distal colon was dissected and collected into ice-cold oxygenated (95% $O_2$ and 5% $CO_2$) Kreb's buffer (116 mM NaCl, 1.3 mM $CaCl_2$, 3.6 mM KCl, 1.4 mM $K_2HPO_4$, 23 mM $HCO_3^-$, and 1.2 mM $MgSO_4$). The colon was opened along the mesenteric border, fecal material was removed, the longitudinal muscle tissue was stripped off, and the tissue was mounted in a horizontal perfusion chamber[34]. Human sigmoid colon biopsies were collected into Krebs' buffer and mounted in a horizontal perfusion chamber as for mouse tissue. Tissue was maintained at 37 °C throughout the duration of each experiment and was perfused basolaterally with Kreb's glucose buffer (Kreb's buffer + 10 mM D-glucose, 5.1 mM sodium L glutamate, and 5.4 mM sodium pyruvate) with Kreb's mannitol buffer (Kreb's buffer + 10 mM D-mannitol, 5.1 mM sodium L glutamate, and 5.4 mM sodium pyruvate) added to the apical chamber. EatA activity on colonic mucus was evaluated by confocal microscopy to assess mucus penetrability, and by measurements of mucus thickness over time. For the penetrability assessment, human and mouse tissue were counterstained with Syto9 (10 μM; Thermo Fisher) for 5 min, and the mucus layer was visualized by the addition of 1 μm FluoSpheres™ carboxylate-modified microspheres (625/645; Thermo Fisher). EatA (10 ng/mL) was added to the apical Krebs' mannitol buffer, and the integrity of the mucus layer was monitored on an upright LSM900 confocal microscope (Carl Zeiss) using a Pan-Apochromat x20/1.0 DIC 75 mm lens (Carl Zeiss). Following the acquisition of an initial scan before the addition of the protease, z-stacks were acquired every 3–5 min for up to 45 min using Zen Blue software (version 3.1; Carl Zeiss). Control analyses were performed on paired mouse or human tissue treated with the non-catalytic EatA variant H134R.

To quantify mucus integrity, beads and tissue surfaces were mapped to isosurfaces using Imaris (version 9.5; Bitplane)[52]. Data regarding the bead position in relation to the tissue surface over time was then extracted and analyzed to generate mucus thickness and normalized bead positional data (Prism version 10.3.1; Graphpad). For the thickness measurements, the mucus surface was visualized by the addition of 10 μm black FluoSpheres™ carboxylate-modified microspheres (Thermo Fisher) to the apical chamber. The thickness of the mucus layer was measured every 15 min for 60 min using a micropipette mounted to a micromanipulator and viewed through a stereomicroscope. The thickness of the mucus layer was obtained by measuring the distance between the mucus surface and the epithelial surface. Following the 60 min measurement, 10 ng/mL EatA was added to the apical chamber, incubated for 15 min followed by one additional measurement of the mucus thickness.

### In silico docking and molecular dynamics

The EatA-DUF domain structure was predicted using AlphaFold2[57]. The cryo-EM structure of MUC2 VWCN-D4-VWC1 (PDB: 7QCN) was modified with Modloop[58] in order to complete the missing loops. The glycosylation sites were modified, ensuring that all of them were occupied with two sugar molecules, two N-acetylglucosamines

(GlcNAcβ1-4GlcNAc) on N-glycosylation sites (Asn[4389], Asn[4453], Asn[4567], Asn[4578], Asn[4703], and Asn[4738]) and galactose and N-acetylgalactosamine (Core-1: Galβ1-3GalNAc) in O-glycosylation sites (Thr[4632] and Thr[4633]). The simplified glycans circumvented the risk of masking interaction surfaces due to their high flexibility, not being considered by the docking programs, while blocking the regions where they are directly attached. The in-silico docking was performed based on shape complementarity using PatchDock[59] and further refined and ranked with FireDock[60]. The EatA structure without signal peptide and β-barrel predicted by AlphaFold2 and a model of MUC2 from the VWCN domain to CK were aligned to the docking models to check their feasibility[20]. The DUF domain was replaced by the full-length EatA in the best-ranked model. The small clashes were solved by a short molecular dynamics simulation (MD). The complex was prepared for MD using the Glycan Reader and Modeler tool from CHARMM-GUI[61]. It was solvated with the TIP3P water model in a cubic simulation box under periodic boundary conditions, neutralized and adjusted to 150 mM NaCl. The generated CHARMM36m force field parameters were used in GROMACS (v2022.4)[62] to perform the energy minimization, equilibration, and production simulations at 303.15 K. The production was simulated for 5 ns. The PTS3 region of MUC2 chain B was modeled in Coot[63], extending it from the VWCN N-terminal residue towards the EatA active site. A mouse PTS3 model was constructed by replacing the human PTS3 sequence in the modeled structure with the corresponding mouse sequence (UniProt: Q80Z19, residue 3760–3812) and subsequently expanded to include all lysine residues. The PTS was glycosylated using GLYCAM, and the resulting model was manually curated in order to remove ring-piercing artifacts. The lateral chains of affected residues were rebuilt, and glycans involved in ring penetrations were removed. A 50 ns MD simulation using CHARMM-GUI and GROMACS. The simulation analysis and figure generation were performed using PyMOL (v2.5) and UCSF Chimera (v1.19). The PTS3 of MUC2 chain A was modeled by applying symmetry operations to the chain B PTS3 only for illustrative purposes.

### Reporting summary

Further information on research design is available in the Nature Portfolio Reporting Summary linked to this article.

## Data availability

The mass spectrometry data have been deposited to the ProteomeXchange Consortium (http://proteomecentral.proteomexchange.org) via the PRIDE partner repository with the dataset identifier PXD068558. All other data generated or analyzed in this study are included in the article and its Supplementary Information. Source data are provided with this paper.

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

## Acknowledgements

This work was supported by the Swedish Research Council grant 2020-02536 (S.v.d.P.) 2023-02474 (G.C.H.), Jeansson's Foundations (S.v.d.P.), Swedish Society for Medical Research (Svenska Sällskapet för Medicinsk Forskning) (S.v.d.P.), Magnus Bergvall's Foundation (S.v.d.P.), Professor Nanna Svartz Foundation 2023-047 (S.v.d.P), Wilhelm and Martina Lundgren's Foundation 2023-SA-4357 (S.v.d.P.), European Research Council (ERC) (101100663, 694181) (G.C.H) GlycoSkin (H2020-ERC) and MSCA-DN (TOP-GUT) (H.H.W), IngaBritt and Arne Lundberg's Foundation (2028-0117) (G.C.H.), The ALF agreement 236501 (G.C.H.), The Michelsen Foundation, The Danish Research Councils, the Lundbeck foundation (H.H.W.). These studies were supported in part by the National Institute of Allergy and Infectious Diseases (NIAID) of the National Institutes of Health (NIH) R01 AI089894, R01 AI126887, and by funding from the Department of Veterans Affairs (5I01BX001469-05) to (J.M.F.). We acknowledge the Mammalian Protein Expression Core Facility for recombinant protein expression and are indebted to the physicians and nurses at the gastroenterology ward of the Sahlgrenska University Hospital. We gratefully acknowledge Yoshiki Narimatsu (University of Copenhagen) for generously providing the MUC2TR1 and MUC2TR2 proteins.

## Author contributions

B.D., S.T-M., G.C.H., and S.v.d.P. conceived and designed the study. J.K.G., F.S., L.A., M.J.G.-B., T.J.V., M.I.N., H.H.W., and J.M.F. contributed with experimental data and interpretation. B.D., S.T.-M., and S.v.d.P. wrote the manuscript, and all authors edited and approved the final version.

## Funding

## Competing interests

The authors declare no competing interests.
