## [Transparent Peer Review file · Nature Communications]

EatA mediated degradation of intestinal mucus is species-specific and driven by MUC2 structural features.

Corresponding Author: Dr Sjoerd van der Post

Version 0:

Reviewer comments:

Reviewer #1

(Remarks to the Author)

This manuscript describes the study of the target proteolytic site for the EatA protease from enterotoxigenic *Escherichia coli* on the MUC2 mucin. Relevant questions are where in MUC2 the protease cuts, whether glycosylation is required for cleavage, and how the protease recognizes MUC2. Overall, the information presented appears reliable and interesting. However, some improvements to the experiments and presentation are necessary. For example, the proper controls (individual proteins alone) are not presented for the gel filtration. Moreover, the story seems relatively straightforward, but the text is overly wordy and poorly organized at points. As noted below, the part describing the modelling of the interaction should be drastically shortened.

Additional comments:

Line 66: "porcess" should be "process"

Line 259: "EatA was only active on the protein C-terminal, resulting in protein fragment of approximately 170kDa (Fig. 1B)." First, no band of size 170 kDa is labeled in Fig. 1B. Second, the significance of this size is not evident from the text. What was the starting size of the protein fragment? What does it mean about the possible cleavage site if a fragment of 170 kDa is generated? What about the complementary fragment resulting from the cleavage?

Line 262: The issue of the reducing vs. non-reducing conditions is not clear. There is the redox status of the reaction during which the proteolysis is taking place, and then there is the redox status of the gel used to assess cleavage. These are two separate things. Whether each of them was oxidizing or reducing should be specified.

Line 267: Here the span of the C-terminal fragment chosen is specified and named (MUC2C-NC), but the first fragment used (line 259) was not specified or named (was it MUC2C?).

Line 273: "only the active form of EatA" This statement is confusing, since no reference to any other form of EatA was presented in the text or used in the previous experiments. The mutant needs to be properly introduced.

Figure 1: The legend says that the lines to the sides of the gel are "color coded," but what is the code? The legend says the lines represent the position of the indicated protein, but the protein is not in fact indicated.

Line 292: "intramolecular" disulfide bonds required for dimerization? Do the authors mean "intermolecular"?

Line 322: "indicating the potential for multiple cleavages" How can the authors distinguish between multiple cleavage sites and ambiguity in site identification? Just because the SK sequence appears multiple times does not mean that all such sites are cleaved. As the authors point out later in the text, other factors in addition to the local sequence determine the actual site of cleavage.

Line 323: "To confirm the cleavage site" The experiment with the peptide does not actually end up confirming the cleavage site, so best not to start the sentence this way. Also, it should be specified that the synthetic peptide is not glycosylated.

Line 327: The topic is the peptide (which is not glycosylated; see previous note), so when the text says, "As this region of the protein is likely to be glycosylated..." the reader is confused for a second until he/she realizes that the authors have jumped to the full protein. In fact, the whole logical progression of this section could be improved. The glycan analysis of MUC2 produced in CHO cells is stuck in the middle. In this reviewer's opinion, it would be more logical to say, "... no cleaved product. We hypothesized that EatA may require O-glycans for activity. We therefore used recombinant GalNAc transferases to add GalNAc to the PTS3 peptide... [describe analysis of the resulting degree and variability of glycosylation]. However, GalNAc addition did not render the peptide cleavable by EatA. To confirm that the degree of glycan modification resembles that of MUC2 produced in CHO cells... Moreover, deglycosylation of MUC2 enhanced proteolysis, indicating that differences in glycosylation between the peptide and native MUC2 are unlikely to account for the differences in cleavage susceptibility. Therefore, another determinant, present in MUC2 but absent from the PTS3 peptide, appears to facilitate cleavage. [start a new paragraph here] EatA has additional domains, outside the catalytic core, which may contribute to substrate recognition. A requirement for binding via these domains to segments of MUC2 that are absent from the PTS3 peptide may explain the inability of EatA to cleave the peptide. To test this hypothesis..."

Line 385: reducing disulfides will likely destroy both secondary and tertiary structure, so best to say, "To determine whether structure is important for substrate recognition, we unfolded..."

Line 389: No need to invoke the PTS3 peptide here. The authors already determined that the peptide lacks important substrate recognition sites.

Line 392: To establish interaction by gel filtration, it is absolutely essential to show the elution profile of each protein alone as well as the mixture of both.

Line 395: The authors cannot conclude anything about whether the interaction is transient or long-lived. It could be that only a fraction of the recombinant EatA is capable of binding and the rest is defective but binds stably (if indeed there is even any binding at all; see previous note).

Line 416: The title of the figure is a bit odd. Yes, EatA and MUC2 domains are involved in substrate recognition, and unfolding of the domains destroys recognition, but domains and folding in general don't "drive" recognition. Also, again the little lines on the sides of the gels are unlabeled and unexplained.

Line 432: "limits" should be "limit"

Regarding the supplementary videos, this reviewer had a difficult time extracting any information from them. Also, at least one of them has repetitive patterns of red blobs appearing and disappearing that look highly artifactual. What is the origin of the uniform spacing seen in these rows of blobs?

Lines 429-486: This paragraph is two pages long. Can it be broken into bite-sized pieces, for example at the beginning of the colon biopsy section?

Line 471: again, "validated" is not the correct word in, "We validated EatA activity on the mouse C-terminal of Muc2..." since there was no activity to validate.

Line 476: "reduction of the EatA band" is not clear. Decrease in intensity of the band?

Line 512: In the opinion of this reviewer, the whole section on the modelling should be drastically shortened and focused. The spatial relationship between putative DUF binding sites on MUC2 and the cleavage site is an interesting question, and the docking is useful for hypothesis generation. However, the docking and related analyses do not "elucidate the molecular mechanism" or justify the mass of detail presented in Fig. 6. One short paragraph and a simple figure discussing/illustrating the spatial arrangements of the relevant parts of the proteins and their proposed interaction modes would be much more effective at this point. (It should also be stated or shown what the differences were between the ranked models.)

Reviewer #2

(Remarks to the Author)

General comments:

The experimental sections of the manuscript are well presented in an easy-to-follow manner. However, the English could use improvement.

1. There are several fragmented sentences in the publication, as such it could use revision.
2. Accordingly, terminus = noun, terminal = adjective; the 2 are not interchangeable.

Introduction:

35-36: The sentence would read better as "In particular, children under the age of five are especially vulnerable to acute symptomatic infections".

38: Spacing surrounding the punctuation following "...stunted growth.."

40: change “and promote” to “..the epithelial surface, enabling the efficient delivery...”
43: Should use “Beyond the adhesins..” rather than “Besides the adhesins”

Comment:

The first paragraph of the introduction talks of the vulnerability and risks to children. As a general question here, what is known about the mucus layer in children? There appears to be a jump in reasoning, without addressing the possible gaps in knowledge.

66: typo: “process”

70: might be better as “..the mucus specific to their host”, rather than “its host”.

76: ad libitum should be in italics.

77: double space following “cycle”

77: “All mouse strains..” rather than “mice strains”

77: “were derived from a C57BL/6N”

78: via in italics.

80: General grammar – when giving further examples, a comma should be used before the examples provided. I.e. “..segment of chromosome 11p15.5, including MUC2 and MUC6.”

98: “serum-free”

104: “cut-off”

105: Are there expected sizes / molecular weights of the constructs?

This will tie into a question later in the manuscript regarding the decoration with more elaborate sugars, but demonstration of a lower molecular weight.

105: “genetically-engineered”

107: “non-catalytic”

113: Should use the long form for JPT, as it only seems to be used twice in the manuscript, and lacks the need to be abbreviated.

121: “For the protease assays, recombinant..” a comma is needed here.

122: rpm is generally not in all caps (RPM)

127: abbreviation MW not introduced.

129-130: Please include the transfer conditions, as mucins are not the easiest to transfer quantitatively.

130: Trans-blot Turbo?

131: This section would benefit from antibody concentrations.

133: “..the band intensities were analyzed” not “was”.

138: Sentence fragmented or information lacking. The “reaction mixture” was applied to the superdex column?

150: N-terminus

161: the word “resolved” is often used in place of “dissolved”.

162: Mass spectrometric analysis

165: (Dr. Maisch), this either needs more information or to be removed.

166: “Peptides.. over 30 minutes” rather than “in 30 minutes”

169: Define HCD.

170: Clarification regarding “subsequent excluded for selection for 20 seconds”

173-176: This could use some organization / clarification.

177: semi trypsin? Please clarify.

178: “..performed using”, rather than “performed towards”

181: missed cleavages

183: The searches “were” performed.

183-184: Needs work/clarification.

184: >300 were “retained / analyzed”?

185: “resolved” versus “dissolved”

195-197: The sentence needs clarification / better punctuation.

General comment for the Ex vivo analysis of EatA mucus degradation: This section could be simplified with subheadings for “Mucus penetrability” and “Mucus integrity quantification”

228: Could you specify the number of sites occupied / involved?

229: N-acetylglucosamine, capital A not needed.

232: “while blocking”

237: “full-length”

Results:

Main concerns regarding the results section:

- 1) Line 259 says digestion of C-terminus results in a fragment of 170kDa. The problem here is that the 170 kDa fragment is present in MUC2C without EatA (image to the left). As such, the statement doesn't make sense, as this does not appear to represent a digestion fragment. The manuscript should make note of the size shift in the MUC2C, rather than the presence of a digestion product.
- 2) Figure 1C. The red bar indicates the presence of the EatA protein. However, in the 3rd lane (MUC2N + EatA), when EatA is added, there is no band detected by Coomassie blue. This is an issue with the manuscript that needs to be resolved.
- 3) In Figure 1A, the limits of the MUC2C-N construct need to be corrected, as I don't believe they span the correct domains (VWC1 should be included in the shaded rectangle under the image).
- 4) Is there an explanation for the differences in the size of the differentially glycosylated PTS domains? Specifically, why

does the addition of a larger glycan result in a reduced size / increased electrophoretic migration? Do you hypothesize an effect on molecular structure or blocking of neighboring glycosylation sites?

5) Regarding the modelling, we're assuming a 1-to-1 interaction in the in vitro studies with expressed mucins, as the cysteines are removed, and thus no dimerization. Is there an explanation why in the modelling the models demonstrate the interaction of EatA with 2 MUC2 chains?

6) This is related to a statement on line 483, indicating that the additional 18 amino acid residues in the rodent Muc2 "will affect the spatial arrangement of the cleavage site region and potential accessibility for the protease." Without direct evidence of this, it should be modified to "will potentially affect...".

7) Otherwise, it would be beneficial to see a modelling/docking interaction of the EatA protein with the rodent Muc2, to demonstrate that the catalytic domain would no longer have access to the cleavage site.

8) Supplemental data Figure 1: Resolution of the image/text is poor, renders the figure unreadable.

259: C-terminus, not C-terminal

261: intermolecular, rather than intramolecular (for dimerization)

263: "resolve" should be switched to "dissociate / digest"

263: N-terminus and C-terminus, not N-/C-terminal

268: C-terminus

271: C-terminus (last time pointing it out, but there are other occurrences in the manuscript)

285-286: This sentence requires clarification in the context of the study. This implies that the enzyme will not be able to cleave native MUC2, while EatA digestion of native MUC2 is demonstrated later in the paper.

288: Terminus

306: Pathogen name incorrect.

303-316: This section introduces a characterized target of SPATEs (IgA), shows similarity of IgA hinge region to MUC2, and goes on to show that EatA does not degrade IgA. What is needed in this section is an explanation of why EatA is active on MUC2 PTS which resembles the hinge region. This needs to be tied back in, as otherwise it leaves the reader confused.

If the IgA hinge region resembles TR-PTS3, and EatA does not digest IgA, then why did you continue to investigate the activity of EatA on the PTS region? This section needs to be re-worked to facilitate easier reader comprehension.

321: Punctuation would help: "TMMP-labelled peptide, SK, was identified", or even to say dipeptide, which would better explain the "SK".

The supplementary tables could be better organized / titled to help the reader.

349: "Generally" not "general".

358: Mass spectrum (as it is singular)

364: spectrum

377: Why are there parentheses around DUF?

388: dependent, not depended

391: "...of and MUC2C-N", remove the "and"

394: Punctuation will help to separate ideas sentences. "...DUF domain was lacking, the interaction..."

396: sensitive "to" dilution.

408: This suggest(s) that..

409: "between the region C-terminal to the cleavage site.." would give better context, rather than "beyond".

412: "...mucin and EatA is essential" not "are essential"

413: For clarity, the DUF interaction domain is C-terminal to the cleavage site, it is not correct to say that it is "beyond", as this lacks context.

Figure 3C: While there is no decrease in size of the principal MUC2 band, there is a reduction in intensity (it seems) and there is an increase in the intensity bands between 150-250kDa.

419: This would represent "decreasing" concentrations, as the amount of enzyme to MUC2 is decreasing from left to right in the image. Or its better just to say "treated at varying protease to substrate ratios".

431: Punctuation: it would read better as "...mesh-like structures, it limits the penetration of the..."

Lack of continuity in the use of "mouse Muc2" and "murine Muc2".

480: "affected species", not "effected species"

485: determinants

501: unpaired Student's t-test?

505: determines species specificity

510: Error in description of the figure. Grey = unaffected (mouse / rat), Red = affected.

522: It should likely be "clashes", rather than "crashes".

556: Please clarify, central region between (not "among") the DUF domain and the C-terminus?

558: suggests

565: got worse?, decreased?

566: remove "so", use "as a result".

570: over time

588 + 590: "shown"

593: "...in size with respect to the central figure"

594: Interacting residues?

Discussion:

Concerns:

1) Lack of discussion regarding interaction with either monomers or dimers / oligomers of MUC2.

613: Remove "resolve", use "degrade" or "dissolve"

618: "localized in a mucin domain"

621: Punctuation for readability. A comma would help this sentence.

628: grammar – glycan structures

631: "alone, suggesting"

632: "Previously, a putative binding domain"

634-635: discussion of binding in the cleft of homodimers (modelling data), but these were not present in the digestion studies. This needs to be addressed here in the discussion.

635-638: Punctuation for readability. Need commas to separate ideas or sections.

Data availability: The identifier given does not work. PDXXXX. Need to replace the place-holder in the text.

Supplementary tables.

When possible, headers in the tables should give more information, such as Band 1, rather than Exp1.

Reviewer #3

(Remarks to the Author)

Version 1:

Reviewer comments:

Reviewer #1

(Remarks to the Author)

The authors have made some improvements to the manuscript, but they seem to have ignored many of the reviewers' requests. Though both reviewers pointed out the general sloppiness of the original version, the revised version has both many of the old typos as well as new typos throughout the added sections. It is frustrating having to go through all the mess again. The lack of care by the authors is a shame, because the results are valuable. I am re-stating my point that the description of the research is still not logical and effective. For example, right after the authors note that they cannot precisely identify the cleavage site because the SK sequence appears multiple times, they write, "To assess the cleavage site..." But then the story goes in the direction of the glycans and never returns to which SK site (if there is actually any selectivity) gets cleaved. It would be better to start a new paragraph when there is a new question, state the question clearly, then describe the experiment done to address it, then report the results, then state the conclusion from those results before starting a new paragraph with a new question. It is not clear why the authors are having a hard time with this.

This reviewer did not think that the description of the modeling was reader-friendly the first time, and, despite claims to the contrary, the authors have not improved the presentation. Yes, some panels were removed from Fig. 6, but other, equally unclear, panels were added. What is the reader supposed to see in panels C and D? For example, this reviewer zoomed in as much as possible and still cannot see the lysine that is referred to in the main text and supposedly labeled in the figure.

Lines 312-367: Again, despite both reviewers pointing out the poor editing of the text in the first round, the authors do not seem to have done much to improve readability. This whole section is one long, unbroken paragraph that is unpleasant to wade through. The scientific questions are straightforward:

- 1) Where is the cleavage site?
- 2) Is glycosylation required?
- 3) Are regions outside the cleavage region itself required for recognition?

These questions were actually addressed in a relatively logical manner experimentally. The text should be re-written so that the logic is clearer to the reader.

Below is a very partial list of remaining or new typos, and additional potential mistakes:

Line 315: should be "between the Fab and Fc domains"

Line 344: should be "GalNAc-transferases"

Line 344: Can the authors please clarify the lengths of the peptides? The text says that a 34-mer was made, but the residue numbers 4328 to 4362 correspond to 35 amino acids and do not match the termini shown in Fig. 2B. Then, what is the "38-mer peptide"?

Line 347: should be "did not result in any"

Line 352: should be "The results highlight the extent of"

Line 366: should be “highly O-glycosylated but is not dependent on glycosylation”

Line 403: “secondary protein structure”?

Line 424: “between the region to the cleavage site”?

Line 425: The CTCK domain is a “cystine” knot, not a “cysteine” knot.

Line 454: A general comment on punctuation: Compound sentences (i.e., two full sentences fused into one with “and” or “but”) should have a comma between the two halves. For example, a comma should be added to yield: “The protease was added to the apical tissue surface, and the thickness of the mucus layer was monitored...” (“The protease” is the subject of the first part of the compound sentence, and “the thickness” is the subject of the second part.) The text should be corrected throughout to apply this rule.

Line 497: should be “addition to humans, pigs, cows, sheep, and dogs are susceptible to ETEC infections, especially”

Line 498: “but not mouse and rat” is awkward tacked onto the end of the sentence. It could be replaced by a new, short sentence saying, “Mouse and rat are not susceptible.”

Line 499: No reason to remind the reader that the rodents are resistant, since this will be stated clearly in the preceding sentence.

Line 532: should be “To gain insight into”

Line 545: should be “the DUF domain inserted into the cavity”

Line 546: should be “the N-terminus of VWCN from chain A” Despite the explicit request by the other reviewer to check “terminus” vs. “terminal” for proper usage throughout the text, the authors did not make this correction.

Line 561: should be “all lysines”

Line 613 what is “(50-10 ns)”? Do the authors mean the panels that are labeled either 10 ns or 50 ns?

Line 629: do the authors mean “degrade” rather than “dissolve”?

Lines 633-637: The first sentence says that mucin domains were considered to be resistant to proteases due to glycosylation, while the second sentence immediately refers to enzymes that cleave mucin domains. Contradiction.

Lines 676-678: How the two parts of this sentence are connected (both logically and grammatically) is unclear.

Lines 678-681: Not a well-formed sentence.

Line 687: There seems to be a period in the middle of the (long, run-on) sentence.

Lines 694-696: This is not a complete sentence.

Lines 698-700: This is a poorly constructed sentence. “limited to” follows “mucus” but actually refers to “proteases”

It is unclear why identification of the cleavage site is necessary for vaccine strategies.

The supplementary figures showing the gel filtration of the two proteins separately are labeled “None reduced” instead of “non-reduced.” Also, the red mark in Figure S2B is above the 100 kDa marker while the protein seems to run below 100 kDa.

The new supplementary figures showing the gel filtration of the two proteins separately (which were added at this reviewer’s request-- thank you) are labeled “None reduced” instead of “non-reduced.” Also, the red mark in Figure S2B is above the 100 kDa marker while the protein seems to run below 100 kDa.

Reviewer #2

(Remarks to the Author)

(please see the attached document regarding formatting colors of the comments)

We greatly appreciate the effort of the authors to update the manuscript addressing our questions and concerns. Nearly all points have been addressed to our satisfaction.

We would like to come back to a response of the authors to look for further clarification.

Response (including reviewer’s question):

"5) Regarding the modelling, we're assuming a 1-to-1 interaction in the in vitro studies with expressed mucins, as the cysteines are removed, and thus no dimerization. Is there an explanation why in the modelling the models demonstrate the interaction of EatA with 2 MUC2 chains?

To clarify we didn't mutate or remove any cysteines in the recombinant proteins included in the study. The C-terminus of MUC2 is always occurs as a dimer stabilized by four intermolecular disulfide bonds, three between the CK domains and one in the VWCN domain (Fig. 1A and PMID:37031240), as demonstrated in figure 1C and 3D. For that reason, all modelling was performed on the dimer. EatA interacts through its MUC2-binding domains with both chains of the MUC2C dimer indicating a 1-to-2 interaction."

It was stated that the cysteines were not mutated or removed, however, in the materials and methods 'lines 95-98 of the updated manuscript':

"with the cysteine residue at position 4786 mutated to an alanine. The mouse Muc2C-N vector was composed similar to the MUC2C-N including the Muc2 sequence between amino acid 3691 and 4248 (UniProt: Q80Z19 v2023_01) with the cysteine residue at position 4231 mutated to an alanine."

It was due to this information in the materials and methods that the reviewers assumed the experimental conditions were assessing a monomeric MUC2 structure, as removing the cysteine would likely impair dimer formation. Could you please clarify the confusion surrounding this? Specifically addressing the purpose of the removal of the cysteine.

Would it be possible to clarify in the manuscript that the EatA or DUF domain interacts with MUC2 dimers. This would remove the confusion encountered by the reviewers in the context of the interaction conditions, i.e. protein:monomer or protein:dimer

Minor comments.

The response to the following comment appears to be unclear or incomplete.

170: Clarification regarding "subsequent excluded for selection for 20 seconds"
Authors' response: "This refers to the ."

The corrected sentence below needs a slight adjustment.
"domain central region C-terminal to the DUF domain", rather than "C-terminally".

556: Please clarify, central region between (not "among") the DUF domain and the C-terminus?
We altered the statement to be more precise about the region of the EatA protein involved in the interaction: "The model reveals a secondary interaction surface that spans around 600 Å² between MUC2 C8 (A) and the EatA passenger domain central region C-terminally of the DUF domain (Extended figure 4C)."

Reviewer #3

(Remarks to the Author)

Degradation of the intestinal mucus layer by the ETEC protease EatA is species-specific and determined by the structure of the MUC2 mucin

Trillo-Muyo, S., *et al.*

Corresponding Author: Sjoerd van der Post

We thank the Editor and Reviewers for their positive review of our manuscript and constructive feedback. Please find below our point-by-point responses to each of the Reviewers' comments. Revisions made to the manuscript and supplemental data are indicated in the respective documents in red.

Reviewer #1:

This manuscript describes the study of the target proteolytic site for the EatA protease from enterotoxigenic *Escherichia coli* on the MUC2 mucin. Relevant questions are where in MUC2 the protease cuts, whether glycosylation is required for cleavage, and how the protease recognizes MUC2. Overall, the information presented appears reliable and interesting. However, some improvements to the experiments and presentation are necessary. For example, the proper controls (individual proteins alone) are not presented for the gel filtration. Moreover, the story seem relatively straightforward, but the text is overly wordy and poorly organized at points. As noted below, the part describing the modelling of the interaction should be drastically shortened.

We thank Reviewer #1 for their thoughtful comments and efforts. In the revised manuscript, we have addressed the concerns raised by improving the clarity and conciseness of the text, adding appropriate controls for the gel filtration analysis and shortening the interaction modeling results section. Below, we provide a point-by-point response to each comment.

Line 66: "porcess" should be "process"

Corrected in the revised manuscript

Line 259: "EatA was only active on the protein C-terminal, resulting in protein fragment of approximately 170kDa (Fig. 1B)." First, no band of size 170 kDa is labeled in Fig. 1B. Second, the significance of this size is not evident from the text. What was the starting size of the protein fragment? What does it mean about the possible cleavage site if a fragment of 170 kDa is generated? What about the complementary fragment resulting from the cleavage?

We agree with the reviewer that the results description lack sufficient detail required for correct data interpretation. In the revised manuscript for clarity, we have replaced the gel electrophoresis image by a repeating the analysis on a gradient gel with a higher resolving power in region of interest, as used for all other analysis throughout the paper. The text has been altered to include the mW of the initial noncleaved molecule and describes the significance of the cleavage product upon EatA treatment, and now reads as following:

Line 262- 266

"When N- and C-terminal regions of MUC2 (hereafter referred to as MUC2N and MUC2C) were treated with EatA in an oxidizing environment and analyzed by SDS-PAGE under reducing conditions, activity was observed only on the C-terminal region (MUC2C). EatA reduced the apparent molecular weight from >250 kDa to ~170 kDa, consistent with the generation of a large proteolytic fragment (Fig. 1B)."

The fragment complementary to the 170 kDa band is predominantly composed of the PTS domains, which are heavily O-glycosylated and therefore not readily visualized by Coomassie

staining. We have made this observation in previous work using the MUC2 protein construct when studying protease activity, see (PMID:1482604 (Fig. 1D) and PMID:23546879 (Fig. 7B).

Updated figure 1B and 1C:

Line 262: The issue of the reducing vs. non-reducing conditions is not clear. There is the redox status of the reaction during which the proteolysis is taking place, and then there is the redox status of the gel used to assess cleavage. These are two separate things. Whether each of them was oxidizing or reducing should be specified.

We clarified the redox status of the proteolysis assay and the gel electrophoresis in the revised manuscript, indicating that only protein separation was performed under either reducing or none-reducing conditions. In addition, we now highlight the oligomeric state in figure 1C and 1D.

Line 267: Here the span of the C-terminal fragment chosen is specified and named (MUC2C-NC), but the first fragment used (line 259) was not specified or named (was it MUC2C?).

We reviewed the paragraph and now included the specific recombinant protein used for the initial experiment (MUC2N and MUC2C). To match the naming scheme in figure 1A.

Line 273: “only the active form of EatA” This statement is confusing, since no reference to any other form of EatA was presented in the text or used in the previous experiments. The mutant needs to be properly introduced.

In the revised manuscript we justify the reason for using the non-catalytic and inhibitor treated protease by including the following sentence.

Line 282 - 286

“To confirm that the observed activity on MUC2C-N was specifically due to EatA and not a contaminating protease, a non-catalytic mutant of EatA was generated by substituting histidine 134 with an arginine. Additionally, the serine protease nature of EatA was validated by treatment with PMSF, a serine protease inhibitor. Neither the mutant EatA nor PMSF-treated EatA was able to degrade MUC2C-N (Fig. 1E)¹³.”

Figure 1: The legend says that the lines to the sides of the gel are “color coded,” but what is the code? The legend says the lines represent the position of the indicated protein, but the protein is not in fact indicated.

We agree with the Reviewer that the description in legend was unclear on how the lines were matched to the proteins used for analysis. Figure legend 1 is now updated with the following appended sentence.

Figure legend 1

“Colored lines underneath the protein names color-match with the lines on the gel image representing the position of the specific protein.”

Line 292: “intramolecular” disulfide bonds required for dimerization? Do the authors mean “intermolecular”?

Corrected in the revised manuscript.

Line 322: “indicating the potential for multiple cleavages” How can the authors distinguish between multiple cleavage sites and ambiguity in site identification? Just because the SK sequence appears multiple times does not mean that all such sites are cleaved. As the authors point out later in the text, other factors in addition to the local sequence determine the actual site of cleavage.

In the revised manuscript we emphasize that the results of the assay demonstrate that EatA acts on this specific region of MUC2, but that based on results we can't determine which of the sites in PTS3 is the actual cleavage site.

Line 335 - 338

“This motif appears three times within the TR-PTS3 region, indicating that EatA acts on this region of MUC2. However, due to the repetitive nature of the sequence, the exact cleavage site could not be determined.”

Line 323: “To confirm the cleavage site” The experiment with the peptide does not actually end up confirming the cleavage site, so best not to start the sentence this way. Also, it should be specified that the synthetic peptide is not glycosylated.

In the revised manuscript we use the term “assess” instead of “confirm” and indicate that the synthetic peptide was non-O-glycosylated.

Line 327: The topic is the peptide (which is not glycosylated; see previous note), so when the text says, “As this region of the protein is likely to be glycosylated...” the reader is confused for a second until he/she realizes that the authors have jumped to the full protein. In fact, the whole logical progression of this section could be improved. The glycan analysis of MUC2 produced in CHO cells is stuck in the middle. In this reviewer’s opinion, it would be more logical to say, “... no cleaved product. We hypothesized that EatA may require O-glycans for activity. We therefore used recombinant GalNAc transferases to add GalNAc to the PTS3 peptide... [describe analysis of the resulting degree and variability of glycosylation]. However, GalNAc addition did not render the peptide cleavable by EatA. To confirm that the degree of glycan modification resembles that of MUC2 produced in CHO cells... Moreover, deglycosylation of MUC2 enhanced proteolysis, indicating that differences in glycosylation between the peptide and native MUC2 are unlikely to account for the differences in cleavage susceptibility. Therefore, another determinant, present in MUC2 but absent from the PTS3 peptide, appears to facilitate cleavage. [start a new paragraph here] EatA has additional domains, outside the catalytic core, which may contribute to substrate recognition. A requirement for binding via these domains to segments of MUC2 that are absent from the PTS3 peptide may explain the inability of EatA to cleave the peptide. To test this hypothesis...”

We thank the reviewer for their insightful feedback on how to improve the readability of this section and the order of data representation. The results section has been updated and now starts with describing the analysis of the glycosylated synthetic peptide followed by the analysis of MUC2 protein.

Line 385: reducing disulfides will likely destroy both secondary and tertiary structure, so best to say, “To determine whether structure is important for substrate recognition, we unfolded...”
 We agree and adapted the sentence stating “*structure*” instead of “*secondary structure*”.

Line 389: No need to invoke the PTS3 peptide here. The authors already determined that the peptide lacks important substrate recognition sites.
 The sentence was removed from revised manuscript.

Line 392: To establish interaction by gel filtration, it is absolutely essential to show the elution profile of each protein alone as well as the mixture of both.
 We included the size exclusion elution profile of the individual proteins to the revised manuscript. The data demonstrates that EatA when combined with MUC2 will elute earlier than when analyzed alone which suggest partial complex formation. The results are included as supplementary figure 2A and B and the following sentence has been added to the main manuscript:

Line 408 - 410
 “The elution profiles of the individually analyzed proteins show that the observed overlap between H134R EatA and MUC2C-N arises from their interaction. (**Extended figure 2A and B**).”

Figure S2A and B:

Line 395: The authors cannot conclude anything about whether the interaction is transient or long-lived. It could be that only a fraction of the recombinant EatA is capable of binding and the rest is defective but binds stably (if indeed there is even any binding at all; see previous note).
 Based on the gel filtration results we can indeed not conclude the nature of the EatA:MUC2 interaction, as emphasized by the reviewer. In the revised manuscript we have removed the claims about the nature of the interaction and the potential reason for why we did not observed a higher proportion of complex formation. The updated sentence reads as follows:

Line 410 - 411
 “The size exclusion chromatography results indicate interaction between protease and substrate.”

Line 416: The title of the figure is a bit odd. Yes, EatA and MUC2 domains are involved in substrate recognition, and unfolding of the domains destroys recognition, but domains and folding in general don’t “drive” recognition. Also, again the little lines on the sides of the gels are

unlabeled and unexplained.

We agree with the reviewer that the title was too specific on aspects not directly responsible for MUC2 recognition and changed the title of the figure legend to:

“MUC2 structural features are critical for recognition by EatA”

To clarify the color coding of each protein on the gel the figure legend was appended with the following sentence.

“Colored lines underneath the protein names color-match with the lines on the gel image representing the position of the specific protein.”

Line 432: “limits” should be “limit”

We corrected the error in the revised manuscript.

Regarding the supplementary videos, this reviewer had a difficult time extracting any information from them. Also, at least one of them has repetitive patterns of red blobs appearing and disappearing that look highly artifactual. What is the origin of the uniform spacing seen in these rows of blobs?

Streaks of beads were observed upon EatA treatment in video 3 and are a technical artifact of the confocal microscopy analysis. Dislodged beads are no longer retained by the mucus layer, and will float away, and can therefore be observed in multiple z-stack during acquisition (see example still below). We agree that the video provided to little experimental details and have now included explanatory text in each video describing the genotype, treatment, and a description of what is visualized in each fluorescent channel.

Lines 429-486: This paragraph is two pages long. Can it be broken into bite-sized pieces, for example at the beginning of the colon biopsy section?

We agree with the reviewer that this section is unnecessary long. We therefor introduced a second result paragraph with the results describing the differences in the cleavage site region with the following title:

Line 485

“Species-Specific variation in EatA cleavage of MUC2/Muc2”

Line 471: again, “validated” is not the correct word in, “We validated EatA activity on the mouse C-terminal of Muc2...” since there was no activity to validate.

In the revised manuscript we use the term “assessed” instead “validated”.

Line 476: “reduction of the EatA band” is not clear. Decrease in intensity of the band?

We agree with the reviewer and changed reduction to “decreased EatA band intensity”.

Line 512: In the opinion of this reviewer, the whole section on the modelling should be drastically shortened and focused. The spatial relationship between putative DUF binding sites on MUC2 and the cleavage site is an interesting question, and the docking is useful for hypothesis generation. However, the docking and related analyses do not “elucidate the molecular mechanism” or justify the mass of detail presented in Fig. 6. One short paragraph and a simple figure discussing/illustrating the spatial arrangements of the relevant parts of the proteins and their proposed interaction modes would be much more effective at this point. (It should also be stated or shown what the differences were between the ranked models.)

Below a point-by-point reply to the comments concerning the results section and figure describing the EatA:MUC2 interaction model.

- We agree with the reviewer and have reduced the level of detail considerably and thereby shorten the length of the paragraph substantially.
- The term “elucidate” was too strong for a hypothesis generating model and replaced with “to gain insight”.
- We decided to move supportive results to the supplement limiting the panels in figure 6 only showing the key aspects of the model (see below). Upon request of reviewer 2 the panel now includes the results of the MD analysis of the mouse Muc2 cleavage site region in the active site of EatA (Fig 6C).
- The presented model was the only model without major clashes between the two proteins, due to these unrealistic close contacts no other models were considered.

Reviewer #2:

General comments:

The experimental sections of the manuscript are well presented in an easy-to-follow manner. However, the English could use improvement.

1. There are several fragmented sentences in the publication, as such it could use revision.

2. Accordingly, terminus = noun, terminal = adjective; the 2 are not interchangeable.

We appreciate Reviewer #2's constructive feedback and the time they dedicated to our manuscript. In the revised manuscript, we have addressed the concerns raised by adding additional gel analysis of MUC2 treated with the EatA proteases to clarify the gel electrophoresis results in figure 1B and 1C. The section describing the interaction model between MUC2 and EatA now includes the results of the analysis of the human MUC2 with the mouse PTS3 regions covering the cleavage site. Demonstrating that the difference in spacing of the lysine residues is likely responsible for the lack of EatA activity on mouse Muc2. In addition, we have implemented the various suggestion made to improve the flow and grammar.

Introduction:

35-36: The sentence would read better as "In particular, children under the age of five are especially vulnerable to acute symptomatic infections".

Adapted

38: Spacing surrounding the punctuation following "...stunted growth.."

corrected

40: change "and promote" to "...the epithelial surface, enabling the efficient delivery..."

Adapted

43: Should use "Beyond the adhesins.." rather than "Besides the adhesins"

corrected

Comment:

The first paragraph of the introduction talks of the vulnerability and risks to children. As a general question here, what is known about the mucus layer in children? There appears to be a jump in reasoning, without addressing the possible gaps in knowledge.

We technically only have this information from rodents where the mucus layer is developing in the first few week postnatal (PMID:40323318).

66: typo: "process"

Corrected

70: might be better as "...the mucus specific to their host", rather than "its host".

Adapted

76: ad libitum should be in italics.

Corrected

77: double space following "cycle"

Corrected

77: "All mouse strains.." rather than "mice strains"

Corrected

77: "were derived from a C57BL/6N"

We appreciate the reviewer's suggestion, as the mice genetic background of all mice in the study was C57BL/6N we prefer to use the term "on" over "from".

78: via in italics.

Via is in generally not italicize in scientific writing. Nature publishing considers it a normal word during typesetting.

80: General grammar – when giving further examples, a comma should be used before the examples provided. I.e. "...segment of chromosome 11p15.5, including MUC2 and MUC6."

Corrected

98: "serum-free"

Corrected

104: "cut-off"

Corrected

105: Are there expected sizes / molecular weights of the constructs?

In the methods section of the revised manuscript we have now included the region of MUC2 covered by the two PTS tandem repeats.

This will tie into a question later in the manuscript regarding the decoration with more elaborate sugars, but demonstration of a lower molecular weight.

The addition of multiple negatively charged sialic acids can alter protein migration, resulting in a substantially lower apparent molecular weight compared to the protein modified with only a T- or Tn antigen.

105: "genetically-engineered"

Corrected

107: "non-catalytic"

Corrected

113: Should use the long form for JPT, as it only seems to be used twice in the manuscript, and lacks the need to be abbreviated.

JPT refers to the company that performed the custom peptide synthesis, in the revised manuscript we written out the full name of the company.

121: "For the protease assays, recombinant.." a comma is needed here.

Corrected

122: rpm is generally not in all caps (RPM)

Corrected to lower case

127: abbreviation MW not introduced.

In the revised manuscript, 'MW' has been spelled out as 'molecular weight'.

129-130: Please include the transfer conditions, as mucins are not the easiest to transfer quantitatively.

In the revised manuscript the details of the blotting conditions are included.

130: Trans-blot Turbo?

Corrected

131: This section would benefit from antibody concentrations.

The antibody concentrations are included in the revised manuscript

133: ".the band intensities were analyzed" not "was".

Corrected

138: Sentence fragmented or information lacking. The "reaction mixture" was applied to the superdex column?

Clarified

150: N-terminus

Corrected

161: the word "resolved" is often used in place of "dissolved".

Corrected

162: Mass spectrometric analysis

Adapted

165: (Dr. Maisch), this either needs more information or to be removed.

Dr. Maisch refers to the manufacturer of the chromatography material

166: "Peptides.. over 30 minutes" rather than "in 30 minutes"

Corrected

169: Define HCD.

Defined

170: Clarification regarding “subsequent excluded for selection for 20 seconds”

This refers to the .

173-176: This could use some organization / clarification.

Rephrased as follows: *“Mascot was used for standard protein identification with the following settings: fixed modification carbamidomethylation (C), variable modification oxidation (M), enzyme trypsin with a maximum number of two missed cleavages, a mass tolerance of 10 ppm for precursor mass and 20 ppm for fragment ions.”*

177: semi trypsin? Please clarify.

This is a general enzyme cleavage rule frequently used in proteomics when identifying peptides for which only one end must follow the trypsin K/R cleavage rule.

178: “..performed using”, rather than “performed towards”

Changed to *“performed against”*

181: missed cleavages

Corrected

183: The searches “were” performed.

Corrected

183-184: Needs work/clarification.

Rephrased as follows: *“The searches were performed against the mucinDB, and a list of 70 common human O-glycans was selected as the set of considered glycan modifications, only glycopeptides with a score. >300 were included.”* Line 184 - 186

184: >300 were “retained / analyzed”?

This is a probability score above which glycopeptides were considered an accurate match.

185: “resolved” versus “dissolved”

Corrected

195-197: The sentence needs clarification / better punctuation.

Rephrased as follows: *“The colon was opened along the mesenteric border, fecal material was removed, the longitudinal muscle tissue was stripped off and the tissue was mounted in a horizontal perfusion chamber.”* Line 197 - 199

General comment for the Ex vivo analysis of EatA mucus degradation: This section could be simplified with subheadings for “Mucus penetrability” and “Mucus integrity quantification”

In the revised manuscript we introduced an indentation before the data analysis section and introduced a heading to break up the section. Line 232

228: Could you specify the number of sites occupied / involved?

The exact modified sites are described in the revised manuscript.

229: N-acetylglucosamine, capital A not needed.

Corrected

232: “while blocking”

Corrected

237: “full-length”

Corrected

Results:

Main concerns regarding the results section:

1) Line 259 says digestion of C-terminus results in a fragment of 170kDa. The problem here is that the 170 kDa fragment is present in MUC2C without EatA (image to the left). As such, the

statement doesn't make sense, as this does not appear to represent a digestion fragment. The manuscript should make note of the size shift in the MUC2C, rather than the presence of a digestion product.

We agree with the reviewer that the results presentation and description lack sufficient detail required for correct data interpretation. In the revised manuscript for clarity, we have replaced the gel electrophoresis image by a repeating the analysis on a gradient gel with a higher resolving power, as used for all other analysis throughout the paper. The text has been altered to include the mW of the initial noncleaved molecule and describes the significance of the cleavage product upon EatA treatment. See updated figure 1B and C below:

2) Figure 1C. The red bar indicates the presence of the EatA protein. However, in the 3rd lane (MUC2N + EatA), when EatA is added, there is no band detected by Coomassie blue. This is an issue with the manuscript that needs to be resolved.

In the new figure 1B and C the EatA is clearly visible in the protease treated conditions.

3) In Figure 1A, the limits of the MUC2C-N construct need to be corrected, as I don't believe they span the correct domains (VWC1 should be included in the shaded rectangle under the image).

We are grateful to the reviewer for pointing out our mistake in the annotation of Figure 1A. The MUC2C-N construct indeed includes the complete VWD domain and almost the complete VWC domain, which is corrected in the revised version of the figure.

4) Is there an explanation for the differences in the size of the differentially glycosylated PTS domains? Specifically, why does the addition of a larger glycan result in a reduced size / increased electrophoretic migration? Do you hypothesize an effect on molecular structure or blocking of neighboring glycosylation sites?

The more extensively O-glycosylated PTS domains are sialylated. The addition of multiple negatively charged sialic acids can alter protein migration, resulting in a substantially lower apparent molecular weight compared to the protein modified with only a T- or Tn antigen. Concerning the blocking of neighboring sites by the proximity of extensive glycosylation, this is unlikely as the synthesis occurs sequentially and the glycosyltransferases responsible for extension are distributed further distally in the Golgi.

5) Regarding the modelling, we're assuming a 1-to-1 interaction in the in vitro studies with expressed mucins, as the cysteines are removed, and thus no dimerization. Is there an explanation why in the modelling the models demonstrate the interaction of EatA with 2 MUC2 chains?

To clarify we didn't mutate or remove any cysteines in the recombinant proteins included in the study. The C-terminus of MUC2 is always occurs as a dimer stabilized by four intermolecular disulfide bonds, three between the CK domains and one in the VWCN domain (Fig. 1A and PMID:37031240), as demonstrated in figure 1C and 3D. For that reason, all modelling was performed on the dimer. EatA interacts through its MUC2-binding domains with both chains of the MUC2C dimer indicating a 1-to-2 interaction.

6) This is related to a statement on line 483, indicating that the additional 18 amino acid residues in the rodent Muc2 “will affect the spatial arrangement of the cleavage site region and potential accessibility for the protease.” Without direct evidence of this, it should be modified to “will potentially affect...”.

In support of our claim, we have now preformed the modelling and molecular dynamics of the mouse PTS3 region within the active site of EatA. As the structure of the mouse MUC2 has not been resolved we replaced the PTS3 domain in the human structure with the mouse sequence. The results of these analysis indicate that three potential cleavage sites are out of range of the active site (Fig. 6D). The site at the correct distance was used for modelling, but loss of interfacial contacts was observed already after ~2 ns, indicating an unstable interaction (Fig. 6D). The simulation indicates that the lysine in the P1' position which is replaced with a glutamic acid in mice dictates the stability of the interaction.

7) Otherwise, it would be beneficial to see a modelling/docking interaction of the EatA protein with the rodent Muc2, to demonstrate that the catalytic domain would no longer have access to the cleavage site.

For detailed answer see point 6

8) Supplemental data Figure 1: Resolution of the image/text is poor, renders the figure unreadable.

In the revised manuscript we have increased the resolution of the supplementary figures.

259: C-terminus, not C-terminal

Corrected

261: intermolecular, rather than intramolecular (for dimerization)

Corrected

263: “resolve” should be switched to “dissociate / digest”

Changed to dissociated

263: N-terminus and C-terminus, not N-/C-terminal

The sentence was adapted on request of reviewer #1 and now reads as follows: “To determine if *EatA* can dissociate dimerized MUC2N or MUC2C the protease assay was performed under oxidizing conditions and analyzed by SDS-PAGE under non-reducing conditions.” Line 282 - 284

268: C-terminus

Corrected

271: C-terminus (last time pointing it out, but there are other occurrences in the manuscript)

Corrected

285-286: This sentence requires clarification in the context of the study. This implies that the enzyme will not be able to cleave native MUC2, while *EatA* digestion of native MUC2 is demonstrated later in the paper.

We agree with the reviewers and removed the statement concerning the disulfide bonds.

“Cleavage in this region results in the degradation of the complex oligomeric mucin structures”

288: Terminus

We adapted the figure panel title to indicate that *EatA* acts on the C-terminal protein region of MUC2

306: Pathogen name incorrect.

Corrected *Haemophilus*

303-316: This section introduces a characterized target of SPATEs (IgA), shows similarity of IgA hinge region to MUC2, and goes on to show that *EatA* does not degrade IgA. What is needed in this section is an explanation of why *EatA* is active on MUC2 PTS which resembles the hinge region. This needs to be tied back in, as otherwise it leaves the reader confused.

If the IgA hinge region resembles TR-PTS3, and *EatA* does not digest IgA, then why did you continue to investigate the activity of *EatA* on the PTS region? This section needs to be re-worked to facilitate easier reader comprehension.

We agree with the reviewer and now included a statement at the end of the paragraph to indicate that likelihood that even though the cleavage site region is similar other regions of *EatA* are specifically tailored for the degradation of MUC2. This will be later in the results section discussed.

Line 324 - 326

“The similarity at the cleavage site, coupled with the lack of EatA activity on IgA1 suggests that other regions of the molecule are important for substrate recognition.”

321: Punctuation would help: “TMMP-labelled peptide, SK, was identified”, or even to say dipeptide, which would better explain the “SK”.

Corrected to “TMMP-labelled dipeptide”

The supplementary tables could be better organized / titled to help the reader.

The tables have been revised and now include a separate tab in the xlxs file with a description of the results presented in each table and a reference to the figure in the main text. In addition, we adapted the same naming scheme for the column as in the manuscript for figure 3F-G.

349: “Generally” not “general”.

Corrected

358: Mass spectrum (as it is singular)

Corrected

364: spectrum

Corrected

377: Why are there parentheses around DUF?

To indicate the abbreviation of domain with unknown function (DUF)

388: dependent, not depended

Corrected

391: “..of and MUC2C-N”, remove the “and”

Corrected

394: Punctuation will help to separate ideas sentences. “...DUF domain was lacking, the interaction...”

Introduced a comma after “*lacking*” as suggested by the reviewer

396: sensitive “to” dilution.

This sentence was revised following the suggestion of reviewer #1, as no evidence was provided to support the claim in the original submission.

408: This suggest(s) that..

Corrected

409: “between the region C-terminal to the cleavage site..” would give better context, rather than “beyond”.

Changed to “*C-terminal to the cleavage site*”

412: “..mucin and EatA is essential” not “are essential”

Corrected

413: For clarity, the DUF interaction domain is C-terminal to the cleavage site, it is not correct to say that it is “beyond”, as this lacks context.

Corrected “*beyond*” to “*C-terminal of*”

Figure 3C: While there is no decrease in size of the principal MUC2 band, there is a reduction in intensity (it seems) and there is an increase in the intensity bands between 150-250kDa.

We interpret these bands as minor impurities that remain present after protein purification by ion exchange chromatography and subsequent size exclusion chromatography. In all other assays EatA activity resulted in a complete absence of the high molecular weight band (~250kDa), and repeated analysis yielded the same results for the unfolded molecule.

419: This would represent “decreasing” concentrations, as the amount of enzyme to MUC2 is decreasing from left to right in the image. Or its better just to say “treated at varying protease to substrate ratios”.

In the revised manuscript the figure legend is adapted according to the reviewers suggestion.

Line 434

“Protein gel electrophoresis of MUC2C treated at varying protease to substrate ratios of EatA or EatA-DUF lacking a domain extending from the helical stalk region.”

431: Punctuation: it would read better as “..mesh-like structures, it limits the penetration of the...”

Adapted

Lack of continuity in the use of “mouse Muc2” and “murine Muc2”.

In the remaining paragraph the distinction between mouse Muc2 and human MUC2 is indicated by the use of lowercase and uppercase.

480: “affected species”, not “effected species”

Corrected

485: determinants

Corrected

501: unpaired Student’s t-test?

Corrected

505: determines species specificity

Corrected

510: Error in description of the figure. Grey = unaffected (mouse / rat), Red = affected.

Corrected

522: It should likely be “clashes”, rather than “crashes”.

Corrected

556: Please clarify, central region between (not “among”) the DUF domain and the C-terminus?

We altered the statement to be more precise about the region of the EatA protein involved in the interaction: *“The model reveals a secondary interaction surface that spans around 600 Å² between MUC2 C8 (A) and the EatA passenger domain central region C-terminally of the DUF domain (Extended figure 4C).”*

558: suggests

Corrected

565: got worse?, decreased?

Corrected to “decreased”

566: remove “so”, use “as a result”.

Corrected to “as a result”

570: over time

We corrected the statement: *“The root mean square deviation (RMSD) stabilized between 15 and 20 ns, indicating that the system reached equilibrium (Extended figure 4D).”*

588 + 590: “shown”

Corrected

593: “..in size with respect to the central figure”

Adapted

594: Interacting residues?

To clarify the terminology used. The residues highlighted in the figure panel are the once on the interface between the surface of the two proteins, therefor we use the word “interfacing residues”.

Discussion:

Concerns:

1) Lack of discussion regarding interaction with either monomers or dimers / oligomers of

MUC2.

The C-terminus of MUC2 always occurs as a dimer supported by four intermolecular disulfide bonds, so we have not considered interaction or degradation with monomers. Our interaction model indicates that the MUC2-binding domain (DUF) interacts with both MUC2 chains of the dimer. Suggesting that it depends on oligomerization for substrate recognition. In the revised manuscript we have added the following sentence to the discussion to clarify this:

Line 651 - 652

“Importantly, proteolysis depends on dimerization of the MUC2C-terminus, as the DUF engages the interface formed by both chains.”

613: Remove “resolve”, use “degrade” or “dissolve”

Corrected to “dissolve”

618: “localized in a mucin domain”

Corrected to “localized in a mucin domain”

621: Punctuation for readability. A comma would help this sentence.

The sentence has been modified to improve readability: *“In contrast to other recently identified O-glycoproteases acting on mucin domains, EatA is not dependent on glycosylation but tolerates glycans in proximity, as supported by molecular dynamics analysis”*

628: grammar – glycan structures

Corrected to “glycan”

631: “alone, suggesting”

Corrected

632: “Previously, a putative binding domain”

Corrected

634-635: discussion of binding in the cleft of homodimers (modelling data), but these were not present in the digestion studies. This needs to be addressed here in the discussion.

The C-terminus of MUC2 always occurs as a dimer. The location of the intermolecular disulfide bonds has been confirmed by Cryo-EM (PMID:37031240), which was used for modeling and data interpretation. This point is hopefully sufficiently clarified in response to reviewer 2 at point 5.

635-638: Punctuation for readability. Need commas to separate ideas or sections.

The punctuation of the sentence was corrected as follow: *“In the absence of the DUF, no cleavage activity was observed, and binding to MUC2 was reduced, highlighting the importance of the DUF domain for initiating interactions with its substrate.”*

Data availability: The identifier given does not work. PXDXXXX. Need to replace the placeholder in the text.

The mass spectrometry data and search results are deposited to PRIDE and can be accessed using the login details below:

Repository: <https://www.ebi.ac.uk/pride/login>

Project accession: PXD068558

Token: crG1mfKvn9zz

Alternatively, reviewer can access the dataset by logging in to the PRIDE website using the following account details:

Username: reviewer_pxd068558@ebi.ac.uk

Password: fzrG4HRdQCnj

RESPONSE TO REVIEWERS

Supplementary tables.

When possible, headers in the tables should give more information, such as Band 1, rather than Exp1.

We have updated the headers in the supplementary table to match the titles used in the manuscript.

Reviewer #3 (Remarks to the Author):

We are grateful for your time and effort.

Degradation of the intestinal mucus layer by the ETEC protease EatA is species-specific and determined by the structure of the MUC2 mucin

Trillo-Muyo, S., *et al.*

Corresponding Author: Sjoerd van der Post

We thank the Editor and Reviewers for their positive review of our manuscript and constructive feedback. Please find below our point-by-point responses to each of the Reviewers' comments. Revisions made to the manuscript and supplemental data are indicated in the respective documents in blue.

Reviewer #1:

The authors have made some improvements to the manuscript, but they seem to have ignored many of the reviewers' requests. Though both reviewers pointed out the general sloppiness of the original version, the revised version has both many of the old typos as well as new typos throughout the added sections. It is frustrating having to go through all the mess again. The lack of care by the authors is a shame, because the results are valuable. I am re-stating my point that the description of the research is still not logical and effective. For example, right after the authors note that they cannot precisely identify the cleavage site because the SK sequence appears multiple times, they write, "To assess the cleavage site..." But then the story goes in the direction of the glycans and never returns to which SK site (if there is actually any selectivity) gets cleaved. It would be better to start a new paragraph when there is a new question, state the question clearly, then describe the experiment done to address it, then report the results, then state the conclusion from those results before starting a new paragraph with a new question. It is not clear why the authors are having a hard time with this.

This reviewer did not think that the description of the modeling was reader-friendly the first time, and, despite claims to the contrary, the authors have not improved the presentation. Yes, some panels were removed from Fig. 6, but other, equally unclear, panels were added. What is the reader supposed to see in panels C and D? For example, this reviewer zoomed in as much as possible and still cannot see the lysine that is referred to in the main text and supposedly labeled in the figure.

We appreciate the reviewer's careful reading. We acknowledge and regret the errors in the previous revision and have amended the text accordingly. The results section describing the cleavage site region identification and the role of O-glycosylation on protease activity is restructured and clarified. In addition, the result section describing the EatA-MUC2 interaction is shortened by excluding multiple figures and restructured to only focus on our key findings.

Lines 312-367: Again, despite both reviewers pointing out the poor editing of the text in the first round, the authors do not seem to have done much to improve readability. This whole section is one long, unbroken paragraph that is unpleasant to wade through. The scientific questions are straightforward:

- 1) Where is the cleavage site?
- 2) Is glycosylation required?
- 3) Are regions outside the cleavage region itself required for recognition?

We apologize that our previous revision did not sufficiently improve this section. We have now substantially edited it to enhance clarity:

- Split the former single paragraph into subsections, each addressing one question and included transition sentences.
- Removed redundancies, shortened sentences, and moved technical detail to keep the narrative concise.

These questions were actually addressed in a relatively logical manner experimentally. The text should be re-written so that the logic is clearer to the reader.

Below is a very partial list of remaining or new typos, and additional potential mistakes:

Line 315: should be “between the Fab and Fc domains”
We changed domain to plural “domains”.

Line 344: should be “GalNAc-transferases”
We have added “transferase” after GalNAc.

Line 344: Can the authors please clarify the lengths of the peptides? The text says that a 34-mer was made, but the residue numbers 4328 to 4362 correspond to 35 amino acids and do not match the termini shown in Fig. 2B. Then, what is the “38-mer peptide”?

The mention of a 38-mer is an error and is corrected to 34-mer in the revised manuscript.
To clarify the length and region of MUC2 that was covered by the recombinant peptide in the various figure panels:

- Figure 2B: Depicting the region of MUC2 that we defined as PTS3 comprising residue 4328 to 4362 is 34 residues in length.
- Figure 2D: The section of the PTS3 region on full length MUC2 for which we identify glycopeptides by mass spectrometry analysis
- Figure 2E: Glycopeptide coverage of the recombinant 34-mer peptide spanning the same region as depicted in 2B (4328 – 4362).
- Figure 5A and D: The PTS3 domain is indicated to be 35 residues long as it includes residue 4363. For sequence comparison between species the PTS3 borders were defined as the region between the last cysteine in the domain before and the first cysteine in the domain after the PTS3 region, justifying the slight ambiguity in length.

Line 347: should be “did not result in any”
We changed results to singular “result”.

Line 352: should be “The results highlight the extent of”
The spelling of extent was corrected.

Line 366: should be “highly O-glycosylated but is not dependent on glycosylation”
The comma after but was removed.

Line 403: “secondary protein structure”?
Included “*secondary*” to protein structure.

Line 424: “between the region to the cleavage site”?
We corrected the sentence as following:

This suggests that in addition to the PTS3, the DUF domain is required for protease interactions between the MUC2 C-terminal region after the cleavage site, which includes the von Willebrand factor CN, D4, C1, C3 domains and cysteine knot.

Line 425: The CTCK domain is a “cystine” knot, not a “cysteine” knot.
Corrected to “cystine” throughout the revised manuscript

Line 454: A general comment on punctuation: Compound sentences (i.e., two full sentences fused into one with “and” or “but”) should have a comma between the two halves. For example, a comma should be added to yield: “The protease was added to the apical tissue surface, and the thickness of the mucus layer was monitored...” (“The protease” is the subject of the first part of the compound sentence, and “the thickness” is the subject of the second part.) The text should be corrected throughout to apply this rule.

We have systematically reviewed the manuscript and corrected punctuation in compound sentences throughout, inserting commas before coordinating conjunctions when they join two independent clauses.

Line 497: should be “addition to humans, pigs, cows, sheep, and dogs are susceptible to ETEC infections, especially”

A comma was introduced after infections.

Line 498: “but not mouse and rat” is awkward tacked onto the end of the sentence. It could be replaced by a new, short sentence saying, “Mouse and rat are not susceptible.”

We moved the statement to the next sentence as suggested.

In addition to humans, pigs, cattle, sheep, and dogs are susceptible to ETEC infection, especially during the neonatal period. By contrast, mice and rats are not.

Line 499: No reason to remind the reader that the rodents are resistant, since this will be stated clearly in the preceding sentence.

We removed the repeated statement in the revised manuscript.

Line 532: should be “To gain insight into”

Corrected to “To gain insight into”

Line 545: should be “the DUF domain inserted into the cavity”

Removed “was”

Line 546: should be “the N-terminus of VWCN from chain A” Despite the explicit request by the other reviewer to check “terminus” vs. “terminal” for proper usage throughout the text, the authors did not make this correction.

Corrected to “terminus”

Line 561: should be “all lysines”

Corrected to “lysines”

Line 613 what is “(50-10 ns)”? Do the authors mean the panels that are labeled either 10 ns or 50 ns?

This indicates the length of the molecular dynamics simulation variation between the human and mouse PTS3 with the active site of EatA. The human PTS3 domain remained in the pocket and the simulation was performed for up to 50ns while for the mouse PTS3 the side chain already left the pocket after less than 10 ns so the simulation time was shortened to 10 ns. We have now clarified this in the figure legend as following:

“EatA active site with human or (E) mouse MUC2 PTS3 detailed interaction before (0 ns) and after (50 ns human PTS3 and 10 ns for mouse PTS3) MD simulation.”

Line 629: do the authors mean “degrade” rather than “dissolve”?

Corrected to “degrade”

Lines 633-637: The first sentence says that mucin domains were considered to be resistant to proteases due to glycosylation, while the second sentence immediately refers to enzymes that cleave mucin domains. Contradiction.

We agree with the reviewer that the section was unclear and the second sentence contradicting. We aimed to acknowledge the existence of O-glycoprotease acting on mucin domains, but lack of evidence that these act on MUC2 and/or can degrade the mucus layer. This is likely limited by the complex and dense O-glycosylation observed in vivo. We updated the section to clarify this as following:

The EatA cleavage site is localized in a mucin domain in the C-terminal part of the protein closely resembling the central mucin domain, which has been considered to be proteolytically resistant due to its extensive O-glycosylation. Various O-glycoproteases acting on mucin domains in other proteins have been identified but none has been demonstrated to actively degrade MUC2, in contrast EatA is not dependent on glycosylation but tolerates glycans in proximity, as supported by molecular dynamics analysis.

Lines 676-678: How the two parts of this sentence are connected (both logically and grammatically) is unclear.

The second sentence has been rephrased in the revised manuscripts as following:

“The colonic mucus layer shares protein composition and O-glycosylation with the small intestine, but is thicker, providing superior visualization of mucus degradation.”

Lines 678-681: Not a well-formed sentence.

See above comment.

Line 687: There seems to be a period in the middle of the (long, run-on) sentence.

We corrected the sentence to improve readability:

“This shared virulence axis among related E. coli pathotypes offers a potential avenue for the rational development of broadly protective vaccine strategies designed to block the initial binding to MUC2 thereby preventing mucus degradation and colonization¹⁴. Progress toward such vaccines requires mapping the regions that mediate substrate recognition, so that inhibitory epitopes can be prioritized for immunogen design.”

Lines 694-696: This is not a complete sentence.

The sentence was rephrased for clarity to:

“We localized the cleavage site to a susceptible region of MUC2 that mediates mucus-gel degradation, and show that interspecies variation in this region underlies host selectivity.”

Lines 698-700: This is a poorly constructed sentence. “limited to” follows “mucus” but actually refers to “proteases”

We reconstructed the sentence according to the reviewers suggestion to clarify our statement.

“The combined results underscore that mucus degradation requires highly specialized proteases, whose expression appears restricted to intestinal pathogens, highlighting a potential strategy to prevent ETEC infection.”

It is unclear why identification of the cleavage site is necessary for vaccine strategies. We agree with the reviewer that identification of the cleavage site is not essential knowledge required for the development of a vaccine. It's more important to identify the domains essential for interaction. In the revised manuscript the part about the cleavage site is removed and now reads as follows:

“Progress toward such vaccines requires mapping the regions that mediate substrate recognition, so that inhibitory epitopes can be prioritized for immunogen design.”

The supplementary figures showing the gel filtration of the two proteins separately are labeled “None reduced” instead of “non-reduced.” Also, the red mark in Figure S2B is above the 100 kDa marker while the protein seems to run below 100 kDa.

We agree with the reviewer and corrected the figure annotation to “nonreduced” in the revised manuscript. In our experience EatA runs around 100 kDa when analysed by protein gel electrophoresis, but agree that the position of the marker was off and corrected this in the revised manuscript.

Reviewer #2 (Remarks to the Author):

(please see the attached document regarding formatting colors of the comments)

We greatly appreciate the effort of the authors to update the manuscript addressing our questions and concerns. Nearly all points have been addressed to our satisfaction.

We would like to come back to a response of the authors to look for further clarification.

Response (including reviewer's question):

We are grateful for your thoughtful review and are glad that nearly all points have been addressed to your satisfaction. Below we restate the remaining question and provide clarification.

“5) Regarding the modelling, we're assuming a 1-to-1 interaction in the in vitro studies with expressed mucins, as the cysteines are removed, and thus no dimerization. Is there an explanation why in the modelling the models demonstrate the interaction of EatA with 2 MUC2 chains?”

To clarify we didn't mutate or remove any cysteines in the recombinant proteins included in the study. The C-terminus of MUC2 is always occurs as a dimer stabilized by four intermolecular disulfide bonds, three between the CK domains and one in the VWCN domain (Fig. 1A and PMID:37031240), as demonstrated in figure 1C and 3D. For that reason, all modelling was performed on the dimer. EatA interacts through its MUC2-binding domains with both chains of the MUC2C dimer indicating a 1-to-2 interaction.”

It was stated that the cysteines were not mutated or removed, however, in the materials and methods ‘lines 95-98 of the updated manuscript’:

“with the cysteine residue at position 4786 mutated to an alanine. The mouse Muc2C-N vector was composed similar to the MUC2C-N including the Muc2 sequence between amino acid 3691

and 4248 (UniProt: Q80Z19 v2023_01) with the cysteine residue at position 4231 mutated to an alanine.”

It was due to this information in the materials and methods that the reviewers assumed the experimental conditions were assessing a monomeric MUC2 structure, as removing the cysteine would likely impair dimer formation. Could you please clarify the confusion surrounding this? Specifically addressing the purpose of the removal of the cysteine.

Would it be possible to clarify in the manuscript that the EatA or DUF domain interacts with MUC2 dimers. This would remove the confusion encountered by the reviewers in the context of the interaction conditions, i.e. protein:monomer or protein:dimer

Our excuses for the confusion. We indeed mutated a cysteine to alanine in two of the recombinant proteins used in the study, but this did not affect the dimer formation. Cysteine 4786 was mutated in the recombinant MUC2C-N and Muc2-N proteins which forms an intramolecular disulfide bond with 4830. The mutation allowed us to generate a truncated version of the protein lacking a large part of the c-terminus with good expression. In figure 1E its shown that the MUC2C-N with the mutated cysteine still forms dimers.

In the revised results section describing the EatA–MUC2 model, we aim to clarify that MUC2 is homodimeric in the proposed complex. We have revised the following sentences accordingly:

Main text:

“To gain insight into the molecular mechanism of EatA-MUC2 recognition and cleavage, we performed docking simulations using a model based on the MUC2 C-terminal homodimer (vWCN–D4–VWC1) cryo-EM structure (PDB: 7QCN) and an AlphaFold2 model of EatA (UniProt:Q84GK0) ²⁴.”

“Consequently, the docking simulations were conducted using the isolated DUF domain first, then extended with the remaining part of EatA to generate a model of the full complex with the MUC2C dimer.”

Figure legend:

“Chains A and B of the MUC2 homodimer are labeled A and B, glycans on chain A are shown in magenta and those on chain B in orange.”

Minor comments.

The response to the following comment appears to be unclear or incomplete.

170: Clarification regarding “subsequent excluded for selection for 20 seconds”

Authors’ response: “This refers to the .”

Our excuses for the incomplete statement in the previous response. The 20 seconds dynamic exclusion means that once a precursor ion is selected for MS/MS, the instrument places that *m/z* (within a set tolerance) on an exclusion list for 20 seconds, preventing it from being reselected for further fragmentation so other precursors can be sampled.

The corrected sentence below needs a slight adjustment.

“domain central region C-terminal to the DUF domain”, rather than “C-terminally”.

556: Please clarify, central region between (not “among”) the DUF domain and the C-terminus?

We altered the statement to be more precise about the region of the EatA protein involved in the interaction: “The model reveals a secondary interaction surface that spans around 600 Å² between MUC2 C8 (A) and the EatA passenger domain central region C-terminally of the DUF domain (Extended figure 4C).”

In the revised manuscript, we removed the figure at the request of Reviewer 1 to clarify the modeling section and maintain focus on the key interaction site.

Reviewer #3 (Remarks to the Author):

We appreciate your time and commitment to this initiative.